# Research Progress on Applications of Polyaniline (PANI) for Electrochemical Energy Storage and Conversion

**DOI:** 10.3390/ma13030548

**Published:** 2020-01-23

**Authors:** Zhihua Li, Liangjun Gong

**Affiliations:** Materials Sciences and Engineering, Central South University, Changsha 41000, Hunan, China

**Keywords:** PANI, electrochemical energy storage and conversion, composites, supercapacitor, rechargeable battery, fuel cell

## Abstract

Conducting polyaniline (PANI) with high conductivity, ease of synthesis, high flexibility, low cost, environmental friendliness and unique redox properties has been extensively applied in electrochemical energy storage and conversion technologies including supercapacitors, rechargeable batteries and fuel cells. Pure PANI exhibits inferior stability as supercapacitive electrode, and can not meet the ever-increasing demand for more stable molecular structure, higher power/energy density and more N-active sites. The combination of PANI and other active materials like carbon materials, metal compounds and other conducting polymers (CPs) can make up for these disadvantages as supercapacitive electrode. As for rechargeable batteries and fuel cells, recent research related to PANI mainly focus on PANI modified composite electrodes and supported composite electrocatalysts respectively. In various PANI based composite structures, PANI usually acts as a conductive layer and network, and the resultant PANI based composites with various unique structures have demonstrated superior electrochemical performance in supercapacitors, rechargeable batteries and fuel cells due to the synergistic effect. Additionally, PANI derived N-doped carbon materials also have been widely used as metal-free electrocatalysts for fuel cells, which is also involved in this review. In the end, we give a brief outline of future advances and research directions on PANI.

## 1. Introduction

With the rapid development of energy, supplying of energy cannot meet the emerging demand [1] due to the increasing energy consumption, which accelerates energy shortage, hence energy storage and conversion play a significant role in overcoming the challenge. To date, different kinds of energy storage and conversion technologies have been developed to deal with the energy crisis. Among them, three kinds of crucial electrochemical energy storage and conversion technologies including supercapacitors, rechargeable batteries and fuel cells [1,2] take the dominance. Figure 1 shows the Ragone plot of supercapacitors, rechargeable batteries and fuel cells.

To a large extent, the performance of electrochemical energy storage and conversion devices are determined by the electrode materials [3]. Carbon species, metal compounds and conducting polymers (CPs) are the main types used as electrode materials. Carbon species that used as electrode materials typically include graphene [4,5,6,7], carbon nanotubes [1], porous nano-carbons [4,8,9] and activated carbon [10] due to their fast charging capabilities and high conductivity [1]. However, their low loading density will result in low energy density, which largely limits their applications for energy storage and conversion. Metal compounds exhibit natural abundance and multielectron redox capability, but they are associated with low conductivity and ease of self-aggregation.

Conducting polymers (CPs) are derived from intrinsically conducting polymers (ICPs) that discovered in 1960 [11]. It caused much attraction from researchers because of the promising properties and potential applications of ICPs since the discovery. CP based devices exhibit higher specific capacitance than double-layer capacitors, moreover, they have faster kinetics than most inorganic batteries, which can narrow the gap between carbon based capacitors and inorganic batteries, indicating the promising potential of CPs in electrochemical energy storage and conversion [12]. Among several common CPs like polyphenyl, polypyrrole, polythiophene, polyphenylacetylene and polyaniline (PANI), PANI generates the most attraction owing to its easier synthesis, lower cost monomer, higher theoretical conductivity (3407 F g^−1^), wider range of working potential window and better stability compared with the other CPs [11]. Therefore, PANI has been a rising superstar in the field of electrochemical energy storage and conversion.

The conductivity of PANI is derived from its unique molecular structure. In 1987, Alan G Mac Diarmid [13] proposed a PANI structural model in which benzene structural units and quinoid structural units co-existed, this structural model has been widely recognized by the scientific community. Diarmid believes that the conductivity of PANI is obtained via doping and de-doping the PANI molecular chain, that is, the PANI molecular chain contains a series of reduced structural units and oxidized structural units, and its structural formula is as follows (Figure 2):

Where: y represents the degree of reduction of PANI, and may also indicate the degree of doping of the molecular chain. When y = 1, it means that PANI is in a fully reduced state (benzene-type structural unit), it is called leucoemeraldine base (LEB); when y = 0, it means that PANI is in a fully oxidized state (quinoid-type structural unit), it is called pernigraniline base (PB); when y is between 0 and 1, indicating a doped state in which an oxidation state and a reduced state co-exist (the benzene structural unit and the quinoid structural unit co-exist). Wherein, when y = 0.5, that is, the dopant is alternately doped in the PANI molecular chain, at this time, the PANI is in an intermediate oxidation state, and the conductivity of the PANI after doping is optimal, the optimal state is called emeraldine base (EB). In general, the PANI that employed as electrode is the mixture of the three states, but the high portion of the EB state is greatly desirable in order to obtain the optimal performance of PANI.

PANI can be synthesized by chemical or electrochemical polymerization of monomer aniline [14]. Various morphologies of PANI can be obtained through different synthesis methods: chemical polymerization usually leads to nanotubes, nanofibers, nanospheres, nanorods, nanoflakes and even nanoflowers, the accurate morphologies are strongly depended on reactive conditions and the species of oxidants; while electrochemical polymerization only leads to nanofibers and films. Different from the chemical polymerization, the morphologies of PANI mainly rely on the natures of the substrates. PANI can combine with other active materials to form a hybrid system. PANI based composites have extra advantages compared with pure PANI, the extra advantages depend on the types of extra active materials, and the composites usually show improved properties. In various PANI based composite structures, PANI usually acts as a conductive layer and network, and the resultant PANI based composites with various unique structures have delivered superior electrochemical performance due to the synergistic effect, which will be introduced in the following sections.

PANI has been widely used in energy storage and conversion, including supercapacitors, rechargeable batteries and fuel cells. When used for supercapacitors, PANI is the active electrode material that acts as a charge carrier during the redox reaction. However, pure PANI often suffers from severe degradation of capacitance and inefficient capacitance contribution during pseudocapacitive process, which can be attributed to the involvement of the swelling, shrinkage and cracking of PANI while doping/dedoping of PANI. Fortunately, PANI with high flexibility can well combine with other active materials like carbon materials, metal compounds and CPs to form PANI based composites with enhanced supercapacitive performance. As for rechargeable batteries, it is a smart strategy to design PANI modified electrode materials. The addition of PANI can not only make up for the disadvantages like cycling instability, low conductivity and structural instability that exist in conventional inorganic electrodes, but also exploit the advantages to contribute synergistic effect. Besides, the hybrid structures also play an important role in electrochemical performance of the composite electrodes. To date now, Pt supported on porous carbon materials is still the most used as an electrocatalyst for fuel cells. However, high cost of Pt severely hinders the commercialization of fuel cells. To enlarge its utilization, various inexpensive metals like Fe, Mo, Mn, Ni and their compounds are chosen to fabricate non-noble metal electrocatalysts, and the alternatives have been demonstrated to address excellent electrocatalytic performance for oxygen oxidation reaction (ORR), hydrogen evolution reaction (HER) and hydrogen oxidation reaction (HOR). For methanol oxidation reaction (MOR), hitherto the main strategy is to design Pt–M (Co, Mo, Ni, Fe, Mn and Ru) alloys catalysts with lower Pt content. PANI with high conductivity, tunable morphologies and high flexibility is an ideal support as metal electrocatalysts. PANI supported metal electrocatalysts show good electrocatalytic activity, furthermore, PANI can be used as support for non-noble metal electrocatalysts, which can cut the cost. Apart from high conductivity, tunable morphologies and high flexibility, PANI supported metal electrocatalysts can effectively suppress the agglomeration through improving the dispersion of the active catalysts. In addition, PANI and its derivatives can be used as the carbon precursor to prepare metal-free non-noble ORR electrocatalysts [15] with enhanced electrocatalytic activity owing to its high N content. Nevertheless, there are fewer reports on catalysts for oxygen evolution reaction (OER) related to PANI, hence it may be another challenging direction for fuel cells.

Several reviews on electrochemical application of PANI have been published in recent years. However, most of them only report exclusive aspect of application in the field of electrochemistry. For example, Snook et al. [12] summarized major conducting polymers (CPs) including PANI, PPy and PTh applied in supercapacitors, as well as their composites with CNTs and inorganic battery. Meng et al. [2] reviewed conducting polymers (CPs) including PANI, PPy and PTh used as supercapacitor electrode materials, and pointed out development directions of CP-based supercapacitors in the future. Luo et al. [16] reported applications of PANI for Li-ion batteries, Li-sulfur batteries and supercapacitors. It is seen from the above mentioned that major work on electrochemical applications of PANI are focused on neither supercapacitors nor batteries, but PANI has been demonstrated to show great potential in various aspects on electrochemistry. Therefore, here we mean a comprehensive review is desirable to fill in the gap. In this review, the wide range of applications of PANI for electrochemical energy storage and conversion technologies including supercapacitors, rechargeable batteries and fuel cells are addressed in detail (as seen in Figure 3), including: (1) PANI based supercapacitor electrodes; (2) PANI modified rechargeable batteries electrodes including lithium-ion batteries, lithium-sulfur batteries and sodium-ion batteries and (3) PANI-based supported metal electrocatalysts and PANI-derived carbon based metal-free electrocatalysts for fuel cells. In the end, we also discuss the future advances and research directions on PANI.

## 2. Applications of PANI for Supercapacitors

Supercapacitors, namely ultracapacitors or electrochemical capacitors, a new energy storage device between conventional capacitors and batteries [17], are considered as the promising electrochemical energy storage/conversion technology due to its high specific power, long cycle lifespan and fast charge/discharge rate [18]. Supercapacitors are generally sorted into two categories [19,20] based on a storage mechanism: an electrostatic double-layer capacitor (EDLC) and pseudocapacitor. EDLCs mainly generate/store energy via adsorbing/desorption on the surface of the electrode by a pure electrostatic charge. Pseudocapacitors, also known as Faraday quasi-capacitors, mainly generate pseudo-capacitance by a reversible redox reaction on the surface and near the surface of pseudo-capacitor active electrode materials (transition metal oxides (TMOs) and conducting polymers (CPs) [21]), thereby realizing energy storage and conversion. The schematics of an electrostatic double-layer capacitor (EDLC) (**a**) and pseudocapacitor are illustrated in Figure 4.

As we know, the performance of supercapacitors strongly depends on the properties of the employed electrode materials. There are three categories of main electrode materials that are adopted in supercapacitors: (1) carbon materials; (2) conducting polymers (CPs) and (3) transition metal oxides (TMOs) [21,22]. Carbon-based materials used as electrode materials of EDLCs have been extensively studied, and it is demonstrated that they exhibit EDLC-type behavior with high power density, low cost and tunable porosity, but they suffer from low energy density [15,23,24,25]. For pseudocapacitors, transition metal oxides (TMOs) and conducting polymers (CPs) are two types of electrode materials that are widely employed. TMOs can display multiple oxidation states [1,15] under low activation energy, but they regrettably suffer from low capacitance, inflexible and instability [15].

CPs, another promising electrode material acknowledged as having high specific capacitance, perfect flexibility, good stability and ease of synthesis, show a fascinating prospect in supercapacitive (pseudocapacitive) electrodes. As mentioned above, PANI has more advantages over the other CPs like ease of synthesis, low cost monomer, high theoretical conductivity (3407 F g^−1^), a wide range of working potential window and good stability, therefore PANI is the most explored material that is used as pseudocapacitive electrodes among CPs [26,27,28,29]. In order to improve its properties, extensive efforts have been made, which are mainly associated with combining PANI with other materials (like carbon-based materials, TMOs). In recent years, substantial work has been devoted into carbon-based materials and TMOs, however reports on PANI are relatively rare. As a kind of novel electrode material with promising properties, it deserves more research and reports because it provides new thought for obtaining supercapacitor electrode with superior performance. In this section, we review the research progress of pure PANI or PANI based composites as pseudocapacitive electrode materials.

### 2.1. Pure PANI

As an outstanding CP with unique characteristics, PANI with multiple redox states has excellent pseudocapacitive performance, therefore many researchers have tried to utilize PANI in supercapacitors since its discovery. An earlier report on its application in supercapacitors appeared in 2001, Florence Fusalba et al. [30] studied electrochemical characterization of PANI. They evaluated stability of PANI-PANI supercapacitor via constant current cycling, a loss of about 60% of the initial charge after 1000 cycles was delivered in the end. They attributed it to high ionic or electronic resistance due to the compact PANI films, and viewed that the morphology of PANI strongly impacted electrochemical properties by influencing ion diffusion into the PANI matrix during redox reaction. More and more researches just demonstrated that PANI morphology plays a crucial role in its electrochemical properties. Chen Hong and his coworkers [31,32] investigated supercapacitive performances of PANI under different morphologies (granular, flake and nanofiber) of PANI films prepared by the pulse galvanostatic method (PGM) and galvanostatic method (GM). The nanofibrous PANI displayed better capacitive performance, which attributed to its lager specific surface and better electronic or ionic conductivity.

Obviously it is important to design a rational nanostructure for the purpose of enhancing the properties of PANI supercapacitive electrodes. In fact, the nanostructure of PANI is strongly determined by the synthesis method, therefore it is crucial to find a rational synthesis method. Liu et al. [33] prepared porous PANI via the in situ aqueous polymerization method, and compared its electrochemical capacitance performance with nonporous PANI. The porous PANI possess smaller and more pores, furthermore the pores presented more random arrangement compared to the nonporous PANI (Figure 5). More fascinatingly, the porous PANI exhibited high specific capacitance as 837 F g^−1^ under the current density of 10 mA g^−1^, much higher than that of the nonporous ones (519 F g^−1^), while its experimental capacitance (1570 F g^−1^) was just about 77% of theoretical value (2027 F g^−1^), indicating that only 77% of PANI makes a contribution to the capacitance ability. Sivakkmar and his coworkers [34] fabricated PANI nanofibers with interfacial polymerization and investigated its properties when used as a supercapacitor electrode. The test showed that the initial specific capacitance (554 F g^−1^) decreased rapidly while cycling, and the value decreased to 57 F g^−1^ after 1000 cycles. Furthermore they found that only a fraction (31%) of the theoretical value of PANI nanofibers was utilized.

In summary, pure PANI used as supercapacitor electrode materials has been investigated a lot, however its unstable cycling and inefficient contribution significantly limit its practical applications for supercapacitors. Hence it is urgent to fabricate various PANI based composites for the purpose of enhancing the electrochemical properties of supercapacitors.

### 2.2. PANI Based Composites

As discussed above, pure PANI may be unsatisfactory to be applied in the supercapacitor electrode because of its instability and limited capacitance contribution. To get over these challenges, researchers are trying to combine PANI with other materials to prepare better electrochemical properties of supercapacitor electrodes. Fortunately, high flexibility of PANI makes it available for PANI to combine with other active materials to form PANI based composites [1], more satisfyingly, these PANI based composites show promising properties when used as supercapacitor electrodes. The PANI based composites can be simply categorized into two types: binary and ternary. Here we will review the recent development and progress of binary and ternary PANI based composites in detail.

#### 2.2.1. PANI Based Binary Composites

It is proved that PANI is compatible with many types of materials like carbon materials, metal compounds and CPs. In the following section, we will discuss the research progress of various PANI based binary composites.

##### PANI/Carbon Binary Composites

Carbon materials as common supercapacitor electrodes have attracted much concern from researchers. Known as acknowledged, carbon materials have lots of outstanding advantages, such as high conductivity, high specific surface area, good stability, perfect electrical properties and so on [35,36,37]. PANI/carbon binary composites are promising to achieve enhanced performance due to excellent electrical properties and great stability of PANI and carbon materials. Many ideal carbon materials including porous carbon, CNTs, graphene and carbon nanofibers have been demonstrated to exhibit outstanding properties when combined with PANI. Porous carbon possess high specific surface area, good stability and large porosity that PANI lacks, which indicates that it is available to prepare PANI/porous carbon composites with improved properties. Chen et al. [35] prepared PANI via electrochemical polymerization, and loaded it on the as-prepared porous carbon electrodes. It was displayed that the initial specific capacitance of the PANI-based capacitor (PC) was as high as 180 F g^−1^, that is, almost double as the bare-carbon capacitor (BC, 92 F g^−1^). After 1000 cycles, the specific capacitance of PC decreased from 180 to 163 F g^−1^, indicating its good cycling stability. More recently, Zhang et al. [36] adopted a facile and economical method to obtain polyaniline/cellulous-derived highly porous activated carbons (PANI/C-ACs) composites. They fabricated C-ACs skeleton via the “selective surface dissolution” (SSD) method [37], in which filter paper was used as the carbon precursor, then PANI nanorods were uniformly grown onto the as-prepared C-ACs skeleton. Figure 6 illustrated the procedure of PANI/C-ACs synthesis. While using it as supercapacitor electrode, it exhibited excellent specific capacitance (765 F g^−1^ at 1 A/g) and high cycling stability (capacitance retention as 91% after 5000 cycles), which was much better than pure PANI. Besides activated carbon (AC), ordered mesoporous carbon (OMC) is also a type of porous carbon with higher specific surface area (1000–2000 m^2^/g) than ordinary porous carbon. As a type of carbon species with unique electrical double-layer capacitance, OMC is desired to combine with PANI with excellent faradaic capacitance, the PANI/OMC composites are promising to largely improve supercapacitor performances. Based on the feasibility, some significant research [38,39,40,41,42] on PANI/OMC composites has been done. In their reports, PANI/OMC composites are successfully synthesized by in situ polymerization or chemical polymerization, and their superiority in supercapacitor electrodes have been illustrated in detail. However different nanostructures of PANI usually exhibit different electrochemical performances, nanofibers, nanorods and nanowhiskers of PANI that deposited onto the surface of OMC showed different specific surface areas and cycling stabilities due to different structure-activity relationships. Herein we take the work of Yan et al. as an example. They [39] synthesized PANI nanowhiskers (PANI-NWs)/ordered mesoporous carbon (CMK-3) composite through chemical oxidative polymerization, and studied its electrochemical performances while used as a supercapacitor electrode. CMK-3, a highly ordered hexagonally mesoporous carbon, owns more superior electrochemical performances than ordinary carbon materials. As expected, the PANI-NWs/CMK-3 showed excellent capacitance retention (90.4% after 1000 cycles) and high electrochemical capacitance attributing to unique vertical arrays of PANI-NWs and ordered framework of CMK-3. As another porous carbon, ordered macroporous carbons are very similar to OMC in many aspects. It can also combine with PANI, and exhibits better electrical double-layer capacitance than PANI/OMC composites owing to its macropore structures that different from OMC [2]. Carbon sphere-type materials are also a kind of important porous carbon materials. Like other porous carbon, they usually possess high specific surface area and pore structures, while their pore structure is very small, so it is more convenient for electrons to transport from the electrolyte to supercapacitor electrode surface, which can contribute to high conductivity and good EDLC performances significantly, whereas it is a popular strategy to combine it with PANI with good Faradaic capacitive properties. Shen et al. [43] fabricated nano-hollow carbon spheres (nano-HCS)/PANI composites via in situ chemical oxidative polymerization. An electrochemical test displayed that the maximum specific capacitance reached 435 F g^−1^, and the capacitance retention was about 60% after 2000 cycles. They declared that the composites are promising for supercapacitor applications.

Carbon nanotubes (CNTs) have been a type of hot materials since the discovery in 1991. Especially in energy storage/conversion, it holds much promise due to its outstanding electrical properties, while its capacitance value is fairly low (generally 40–80 F g^−1^) attributing to its small specific surface area [44], so it is urgent to improve its capacitance properties. Several reversible oxidation states of PANI endows it with the feasibility that enhancing the properties of CNTs through fabricating PANI/CNTs composites. Khomenko et al. [44] firstly employed PANI/multi-wall carbon nanotubes (MWCNTs) composites in supercapacitor electrodes. They obtained the composites via chemical oxidative polymerization method, the PANI deposited onto the surface of the as-prepared MWCNTs during the polymerization. The composites as a positive electrode exhibited a specific capacitance of 320 F g^−1^ (almost eight times as that of MWCNTs) and a loss of about 8% of initial capacitance after 50 cycles. Right after Khomenko, Deng et al. [45] synthesized CNTs/PANI nanocomposite via the deposition of PANI on the surface of CNTs, which is a facile and cheap method as they claimed. In the nanocomposite, CNTs worked as the skeleton so as to increase the specific surface area of deposited PANI, which served as the skin. The unique skeleton/skin structure and excellent pseudocapacitance of uniformly coated PANI largely enhanced the specific capacitance of the composite where the value reached 183 F g^−1^, much higher than the CNTs as 47 F g^−1^. Yang’s group [46] prepared nitrogen-containing CNTs/PANI nanocomposite with tunable morphologies by high temperature treatment. When the composite was used as the supercapacitor electrode in KOH solution, they showed high specific capacitance (163 F g^−1^ at 700 °C and 0.1 A g^−1^) and good cycling stability. More recently, Wang and his coworkers [47] reported a novel method to fabricate flexible ultrathin all-solid-state supercapacitors with excellent electrochemical performances. They synthesized a single wall carbon nanotubes (SWCNTs)/PANI electrode film and a PVA/H_3_PO_4_ electrolyte through spray-printing and spin-coating methods respectively. The SWCNT/PANI electrode presented a considerable initial specific capacitance of 355.5 F g^−1^ when the mass ratio of SWCNT:PANI is 1:1. Moreover, its capacitance retention reached 87.2% of its initial specific capacitance after 5000 cycles, whose electrochemical properties are superior to the former [44,45,46], the author held the view that the flexible ultrathin all-solid-state supercapacitor is promising to pave the way for advanced applications of energy storage.

Graphene, a popularly studied material with extraordinary electrical, mechanical and thermal properties, has attracted extensive concern from researchers. Furthermore, graphene has been the hot materials served as supercapacitor electrodes due to its excellent conductivity and considerable theoretical surface area (2630 m^2^/g) [48,49,50]. Additionally, graphene possesses high structural stability that PANI lacks, hence it is a fairly ideal material to combine with PANI for the purpose of optimizing supercapacitor performances. Wu et al. [48] prepared a free-standing, flexible chemically converted graphene (CCG)/polyaniline nanofibers (PANI-NFs) composite film with a layered structure in the stable aqueous dispersions by vacuum filtration of the mixed dispersions. The as-prepared composite film was with high conductivity of 550 S/m, high specific capacitance of 210 F g^−1^ at 0.3 A g^−1^ and stable cycling property (94% is maintained after 1000 cycles), much better than that of pure PANI-NFs, indicating that the addition of graphene displays a noticeable promotion on PANI. Zhang et al. [49] obtained a graphene/PANI nanofiber composite by in situ polymerization in the presence of graphene oxide (GO). They found that composites’ electrochemical properties were at different conditions when the content of GO changed, moreover, the highest specific capacitance of 480 F/g at 0.1 A/g was achieved when GO content (mass ratio) was 80%, and 70% of initial capacitance was retained over 1000 cycles. They explained that the individual graphene sheet is easy to agglomerate while at service, but it can be improved by the addition of GO that is hard to agglomerate. Generally, graphene applied in supercapacitors is synthesized through the Hummer’s method or the modified Hummer’s method on account of its low cost and high yields [1], then extracts the graphene oxide (GO) and reduced GO (RGO), two important graphene derivatives, and PANI/GO (or RGO) composites have been extensively studied in recent years. Xu et al. [50] reported a facile method to fabricate the GO/PANI nanocomposite via in situ polymerization with the assistance of supercritical carbon dioxide (SC CO_2_). Figure 7 demonstrates the synthesis process. As we can see in Figure 7, polymerization occurred under CO_2_ atmosphere, which can effectively promote the dispersion of the aniline monomer. Tunable morphology of the PANI/GO nanocomposite can be achieved through controlling the concentration of aniline during polymerization. When the concentration was 0.1 M, the nanocomposite achieved a high specific capacitance of 425 F/g at 0.2 A/g and still retained 83% of initial capacitance over 500 cycles, superior to individual PANI at the same condition. As the group illustrated, the synergistic effect between the nanosized PANI and GO with high specific surface area leads to the excellent electrochemical properties. Wang and coworkers [51] designed a soft chemical route to prepare PANI doped with GO sheet. The obtained nanocomposite possess high conductivity of 10 S/m at 22 °C and high specific capacitance of 531 F/g at 0.2 A/g, better than that of pure PANI (216 F/g). However, PANI/GO composites’ capacitance is still not so high due to the intrinsic surface nature of GO [52], which might influence the efficient electron transfer, that, in turn, could hinder its usage in supercapacitors. Reduced GO (RGO) can significantly increase capacitance and conductivity because of a decrease of oxygen-containing functional groups and recovery of the perfect graphene structure, and it is strongly demonstrated by Luo’s work [52]. In their work, RGO was compounded with PANI by the following strategy: firstly, GO was reduced by glucose and ammonia; then PANI was uniformly in situ polymerized onto the as-prepared RGO nanosheets. The reduction degree of GO was measured by the reduction time. Conductivity and capacitance of RGO was increased with an increase of reduction time due to decrease of O content on the surface of GO. Particularly, the optimum PANI/RGO supercapacitive performances were achieved where specific capacitance was as high as 1045 F/g and a high retention of 97% after 1000 cycles occurred at a reduction time of 1 h.

Carbon nanofibers also attract some attention from researchers. More recently, Meltem’s group [53,54] reported on the preparation of free-standing flexible PANI/carbon nanofiber electrodes by the sol–gel and electrospinning method. Compared with an individual carbon nanofiber electrode, the hybrid electrode was with high specific capacitance of 234 F/g and great cycling stability with a capacitance retention of 90% after 1000 cycles, along with high energy density of 32 Wh/kg at a power density of 500 W/kg benefiting from excellent pseudocapacitive properties of PANI coating onto the carbon nanofiber.

In conclusion, it is a promising strategy to prepare PANI/carbon binary composites to optimize the electrochemical performances. Morphology and structure of the composites have a key impact on the electrochemical properties of PANI based electrodes, hence it is crucial to design a rational way to achieve the targets. In general, the morphology of nanofiber, nanowhisker or free-standing flexible 3D-structure works better than other morphologies and structures. However, the limited improvement of specific capacitance is still the problem of carbon materials [2], fortunately, metal compounds (especially metal oxides) make up for the shortage, therefore, PANI/metal compound deserves extensive investigations in supercapacitors.

##### PANI/Metal Compounds Binary Composites

Except for carbon materials, metal compounds also have been popularly studied due to their large capacitance ability and excellent stability. However, their conductivity is fairly low, thus it is necessary to involve in the assistance of CPs (especially PANI) with high conductivity. In recent years, metal oxides, metal chlorides, metal sulfides and metal nitrides are commonly used to combine with PANI as supercapacitor electrodes for improving the electrochemical performance.

Metal oxides, especially transition metal oxides, recognized as a candidate in the field of a supercapacitor benefiting from their extraordinary stability and outstanding electron storage ability, are most used among metal compounds. However their inferior conductivity and poor resistance to acid greatly hinder their application. In order to overcome the limitation, many researchers fabricate CPs/metal oxides hybrids with excellent conductivity, and PANI/metal oxides are more commonly reported. In the field of PANI/metal oxides hybrids, MnO_2_ is most widely studied because of its lower cost, available in abundance, excellent electrical and capacitive behaviors, along with environmental friendliness [55,56,57,58,59]. Liu et al. [60] reported a facile one-step method to electrochemically synthesize the PANI/MnO_2_ composite via pulse electrodeposition. Wherein, MnO_2_ particles were uniformly dispersed onto the surface of PANI nanorods. The prepared composite possesses a high specific capacitance of 810 F/g at 0.5 A/g and a capacitance retention of 86.3% after 1000 cycles, much higher than pure PANI, reflecting that the addition of MnO_2_ has synergistic effects between the involved materials. Novel work was done by Huo’s group [61]. Firstly, they prepared PANI/MnO_2_ nanofibers through interfacial chemical polymerization. During the synthesis, 4-amino-thiophenol (4-ATP) acted as the structure-directing agent on the Au substrate; then the nanofiber composite with a size of 30 nm transferred into the microsphere by self-assembly. The PANI/MnO_2_ nanofiber microsphere electrode obtained preferable specific capacitance of 765 F/g at 1 mA/cm^2^ in 1 M Na_2_SO_4_ solution, and high cycling stability (a capacitance decrease of just 14.9% after 400 cycles), which confirm that the hybrid might be a promising supercapacitor electrode material. Ran and coworkers [62] fabricated a nano-PANI@MnO_2_ hybrid with a tubular shape via a surface initiated polymerization technique. The composite electrode showed good electrochemical performances including a specific capacitance of 386 F/g in 1 M NaNO_3_ electrolyte with the potential window range from 0 to 0.6 V, excellent stability (a retention of 79.5% after 800 cycles) as well as perfect EDLC performance. Compared with individual PANI or MnO_2_, the hybrid’s performance significantly improved, which can ascribe to a synergistic effect between the two compositions_,_ Xie et al. [63] revealed that the structure of MnO_2_ could decide the properties of PANI/MnO_2_ composite electrodes in their study. Four crystal structures of MnO_2_ were fabricated in their work, they are α-MnO_2_, β-MnO_2_, γ-MnO_2_ and δ-MnO_2_ respectively. Wherein, γ-MnO_2_ exhibited the best effect through comparing the electrochemical performance of the four crystallographic structures, thus it was elected to synthesis γ-MnO_2_/PANI nanocomposite electrode through the in situ polymerization method. The γ-MnO_2_/PANI modified electrode showed enhanced electrochemical performance (493 F/g at 0.5 A/g and 95% capacitance retention after 1000 cycles) than individual γ-MnO_2_ or PANI while the mass ratio of γ-MnO_2_ to PANI was 1:5. The author further explained that three crucial factors lead to that: high energy density and excellent cycling stability of γ-MnO_2_, high conductivity of PANI, along with interconnected reticular structure between the two compositions.

Besides MnO_2_, PANI/other metal oxides composite electrodes’ properties are also studied extensively and a lot of progress has been made in recent years. For instance, Ates et al. [64] compared electrochemical performance of PANI/CuO, PPy/CuO and PEDOT/CuO nanocomposite films that electrochemically fabricated on glassy carbon electrode. Results displayed that PANI/CuO has advantages over the others with a highest specific capacitance of 286.35 F/g at 20 mV/s, while the highest specific capacitance value of PPy/CuO and PEDOT/CuO was 20.78 F/g and 198.89 F/g at 5 mV/s respectively. It can be obviously included that PANI/CuO was the most ideal candidate for supercapacitor electrode materials. Giri’s group [65] successfully obtained RuO_2_/PANI composite via in situ oxidative polymerization and investigated the effect on electrochemical performance while doping with Ru (III). As the electrochemical characterizations showed, a specific capacitance of 425 F/g had been achieved after addition of RuO_2_, while the pure PANI was 160 F/g, meanwhile, the composite’s cycling stability also largely increased, reflecting that RuO_2_ brought about a synergistic effect. More recently, Prasankumar and coworkers [66] prepared the PANI/Fe_3_O_4_ composite via in situ polymerization in the presence of microwave obtained Fe_3_O_4_ nanoparticles. The prepared composite exhibited high specific capacitance of 572 F/g at 0.5 A/g and pronounced long-term cycling stability (82% capacitance retention over 5000 cycles). The group viewed that it is of great potential to be applied as efficient supercapacitor electrodes. Wang’s group [67] successfully prepared the SnO_2_@PANI nanocomposite as below (schematically illustrated in Figure 8): To start, SnO was obtained by ultrasonication in the presence of ethanolamine (ETA). Then followed by oxidation to SnO_2_ by in situ oxidation and the polymerization of aniline. The optimum SnO_2_@PANI composite (18.73 wt% PANI) possesses a specific capacitance as high as 335.5 F/g at 0.1 A/g, excellent rate capability (108.8 F/g at 40 A/g) and perfect cycling stability (no capacitance decay after 1000 cycles), superior to pure PANI, which benefits from the synergistic effect where the enhanced stability is derived from SnO_2_ and the enhanced capacitance is derived from PANI.

Apart from metal oxides, other metal compounds like metal chlorides, metal sulfides and metal nitrides also are appropriate to be exploited as electrode materials for supercapacitor. Compared with metal oxides, they are more stable in acidic electrolyte, but their electrical conductivity and electrochemical performance are not so well as metal oxides. Dhibar et al. [68] synthesized PANI/CuCl_2_ composites with various doping levels of CuCl_2_ (1, 2, 3 and 4 wt%) through in situ oxidative polymerization, using APS (Ammonium persulphate) as an oxidant in the HCl medium. Electrochemical measures showed that PANI/CuCl_2_ (2 wt%) exhibited the maximum specific capacitance of 626 F/g at 10 mV/s, meanwhile the composite electrode was with the maximum power density (8158.5 W/kg at 200 mV/s) and maximum energy density (222.57 Wh/kg at 10 mV/s), which indicate that the content of CuCl_2_ has an important effect on composites’ electrochemical performance. More lately, Zhang et al. [69] developed a template-assisted technique to synthesize MoS_2_/PANI hollow microspheres. PANI was deposited onto the surface of hollow MoS_2_ microspheres. The formation process of MoS_2_/PANI hollow microspheres is vividly displayed in Figure 9. They pointed out that hollow MoS_2_ microsphere structure provide a large number of ion channels and large surface area, which is in the flavor of ion transport in an electrolyte. When the mass ratio of MoS_2_:PANI was 1:2, the electrode possess maximum specific capacitance of 364 F/g at 5 mV/s and capacitance retention of 84.3% after 8000 cycles at 10 A/g. From Dhibar and Zhang’s work, we can see that the content of metal compound has a crucial effect on composites’ electrochemical performance, and it is not a linear relationship between the content and electrochemical performance, the best electrochemical performance corresponds to a certain value of content. Xia et al. [70] designed a PANI/TiN core–shell nanowire arrays (NWAs) structure by a facile electrodeposition technique. The PANI/TiN NWAs electrode was with a very high specific capacitance of 1064 F/g at 1 A/g and a stable capacitance retention of 95% after 200 cycles benefiting from high electronic conductivity and capacity storage derived from the core–shell NWAs structure.

##### PANI/CPs Binary Composites

PANI can combine with other CPs to form co-polymers. Due to the intrinsic electrical conductivity in CPs, along with their excellent pseudocapacitive performance, these co-polymers usually possess enhanced supercapacitive performance derived from a synergistic effect. Additionally, the co-polymers are desirable because of their low-cost synthesis, high energy storage capacity, high yields and environmental friendliness [71,72,73,74,75].

In recent years, more and more investigations on PANI/CPs composites are done, and more and more satisfactory properties (especially electrochemical properties) on them are explored out, obviously they might hold much promise for serving for supercapacitors. Zhang et al. [75] designed novel PANI/PPy double-walled nanotube arrays (DNTAs; as schematically illustrated in Figure 10). The fabricated hybrid DNTAs were used as the working electrode to study its electrochemical properties. The studies showed that the PANI/PPy DNTAs exhibited a high specific capacitance of 693 F/g at 5 mV/s, which was much higher than PPy DNTAs (250 F/g at same condition), outstanding rate capability and excellent long-term cycling stability (7.6% capacitance loss after 1000 cycles). They highlighted that both of PANI and PPy made a contribution to improved electrochemical performance. Very recently, Yang et al. [76] co-polymerized PANI and PEDOT by a molecular bridge provided by phytic acid. The PANI/PEDOT co-polymer hydrogel was with a 3D-network structure of PEDOT sheets where PANI was inlaid. While used as a supercapacitor electrode, it showed outstanding electrochemical performance and highly enhanced mechanical properties ascribing to a synergistic effect and unique molecular interactions between PANI and PEDOT.

##### Other PANI Based Binary Composites

Apart from carbon, metal compounds and CPs, some composites that consist of PANI and other materials also have been found to show improved properties recently, including PANI/pure metal, PANI/other organic materials and so on. Tang et al. [77] developed a green strategy to obtain a PANI/Ag composite where Ag nanoparticles were uniformly dispersed onto the surface of PANI via reduction Ag substrate with the assistance of vitamin C. The composite achieved the maximum specific capacitance of 553 F/g at 1 A/g, which was much higher than pure PANI (316 F/g at 1 A/g), attributing to the synergistic effect between PANI and Ag. Its cycling stability (about 90% capacitance retention after 1000 cycles) and rate capability were in good condition. Moreover, it exhibited a high electrical conductivity of 215.8 S/m, it is because Ag nanoparticles promote charge transfer between the active components, thus leading to enhanced specific capacitance and conductivity. Chen et al. [78] designed a cabbage-like PANI/hydroquinone composite microsphere through in situ polymerization. Electrochemical investigations demonstrated that the nanocomposite exhibited great electrochemical properties, that is, a high specific capacitance of 126 F/g at 5 mV/s and 85.1% capacitance retention after 500 cycles at 1 A/g. As the author viewed, PANI provided electronic conductive channels for the hydroquinone and hydroquinone that act as a pseudocapacitance component.

#### 2.2.2. PANI Based Ternary Composites

As reviewed above, PANI based binary composites have made a lot of contribution to supercapacitor electrodes, however, the optimum electrochemical properties such as conductivity, specific capacitance and cycling stability still can not be achieved fully [2]. Currently, in order to reach the utmost electrochemical performance levels, more and more attentions are focused on ternary composites.

PANI/metal oxides/carbon materials ternary composites are most applied in supercapacitors since CPs, metal oxides and carbon materials are the most outstanding active materials for supercapacitor electrodes. Sankar and Selvan [79] designed a ternary hybrid supercapacitor electrode where the obtained MnFe_2_O_4_ nanoparticles were dispersed on the flexible graphene sheets via the hydrothermal method and were wrapped with PANI via the in situ chemical polymerization method. In the ternary MnFe_2_O_4_/graphene/PANI hybrid structure, the MnFe_2_O_4_ and PANI functioned as a spacer that avoids reattachment between the flexible graphene sheets, and PANI and graphene functioned as a conductive network that promotes ion transfer from an electrolyte into an electrode. As expected, the composite exhibited satisfactory electrochemical performance, including a high specific capacitance of 241 F/g at 0.5 mA/cm^2^ (7.5 times higher than MnFe_2_O_4_) and perfect cycling stability (100% capacitance retention after 5000 cycles). As we can see, the ternary possesses more pronounced electrochemical performance (especially cycling stability) than the binary composites, it mainly attributes to the stronger synergistic effect between the three constituents than the binary ones. Fu’s group [80] developed a facile electro-polymerization strategy to fabricate a 3D polystyrene microsphere-reduced graphene oxide/MnO_2_/PANI (3DrGN-MnO_2_-PANI) coaxial arrays composite. During the fabrication process of the 3DrGN-MnO_2_-PANI composite, the polystyrene (PS) was inserted between the rGN templates, which enlarged the specific surface area of the rGN; PANI and MnO_2_ dispersed onto the rGN templates with arrays and nanoflake structure respectively, which shortened the ion diffusion path, enlarged interfacial area and fastens electrical pathways. As expected, the ternary composite film electrode showed a high specific capacitance of 1181 F/g at 1 A/g and good cycling stability with 89.1% capacitance retention after 1000 cycles at 20 A/g. The enhanced electrochemical performance proves that the 3DrGN-MnO_2_-PANI would play a significant role in energy storage systems. It is very recently that Jeyaranjan and coworkers [81] reported a highly scalable ternary porous hierarchical PANI/RGO/CeO_2_ hybrid microsphere prepared by a spray drying method. The obtained ternary microsphere was with a high specific capacitance of 684 F/g, good rate capability and excellent long-term cycling stability with 92% capacitance retention after 6000 cycles. The improved electrochemical properties attribute to the functional and synergistic effect between the three constituents.

PANI/metal oxides/carbon materials are not the exclusive PANI based ternary composites for supercapacitor electrodes. In fact, the other components except PANI need to make up for the shortages of PANI for the purpose of fulfilling the maximum synergistic effect level as possible. As for PANI, the disadvantages including the poor stability, not so high specific surface area, relatively low electronic conductivity and ease of agglomeration are urgent to be enhanced in the PANI based ternary systems, thus the other two components that can act as optimizing these disadvantages of PANI are suitable to form PANI based ternary systems. For example, Kim et al. [82] synthesized ternary Ag/MnO_2_/PANI nanocomposite films for supercapacitor electrodes through a novel pulsed potential electro-deposition strategy. In the composite, Ag provides a high electronic conductivity and fast ion transfer, furthermore, Ag and MnO_2_ shaped as uniform vermicular morphology while the pure PANI shaped as agglomerated vermicular-like structure, both of which improve the electrochemical performance of the composite electrode. As a result, the ternary composite exhibited much better electrochemical performance than pure PANI film with a high calculated specific capacitance of 621 F/g and 800 F/g from CV and CD respectively, as well as good stability (83% capacitance retention after 750 cycles). Lu et al. [83] prepared PANI@TiO_2_/Ti_3_C_2_T_x_ ternary composite with a hierarchical structure via a hydrothermal treatment with addition of the in situ polymerization process. As new active materials, the layered Ti_3_C_2_T_x_ provided a high specific surface area and a nice framework that is beneficial to ion transfer, and PANI nanoflakes and TiO_2_ nanoparticles can increase the active materials’ surface area. As electrochemical measures showed, the nanocomposite a remarkable stability of 94% capacitance retention after 8000 cycles at 1 A/g. They explained that the high specific capacitance of 188.3 F/g at 10 mV/s, while TiO_2_/Ti_3_C_2_T_x_ was about half of that, and a hierarchical architecture and the synergistic effect of combining PANI nanoflakes with TiO_2_/Ti_3_C_2_T_x_ composite resulted in the enhanced electrochemical performance. More recently, Zhu and coworkers [84] developed an effective method to construct hierarchical ZnO@metal-organic framework (MOF)@PANI core–shell nanorod arrays on carbon cloth (CC) for a supercapacitor electrode. The schematic construction of ZnO@MOF@PANI nanoarrays on CC is vividly illustrated in Figure 11. As rising porous materials, MOF is associated with considerable high specific surface area and fast electron and ion transfer, and ZnO is related with excellent stability, hence the ZnO@MOF@PANI might be a good combination for a supercapacitor electrode. In the composite architecture, ZnO nanorods acted as the core that supports the MOF@PANI shell. Results showed that the ternary composite possesses a high specific capacitance of 340.7 F/g at 1.0 A/g, good rate capability with 84.3% capacitance retention from 1.0 to 10 A/g and long-term cycling stability of 82.5% capacitance retention after 5000 cycles at 2.0 A/g. The enhanced electrochemical properties benefited from the unique hierarchical core–shell architecture and the synergistic effect of the three components as the author viewed.

Table 1 presents the preparation method and electrochemical performance of some typical PANI based supercapacitor electrode materials.

In summary, PANI based ternary composites show great potential in supercapacitors. Furthermore, the electrochemical performance is deeply influenced by the nature of the components that combined with PANI and the structure or morphology of the composites, thus it is important to look for materials with a synergistic effect, and to design a suitable nano-structure or morphology, then result in extensive investigations. Obviously, study on PANI based ternary electrode materials has become a hot direction in the field of energy storage and conversion.

## 3. Applications of PANI for Rechargeable Batteries

As another important electrochemical energy storage and conversion device, rechargeable batteries, also called secondary batteries, have been extensively used owning to its high energy density, good portability, low cost, safe and excellent stability. Just like supercapacitors, their electrochemical performances are greatly related with electrode properties. Conventional electrode materials for rechargeable batteries are mainly metal or metallic derivatives, but they usually suffer from poor stability, inferior conductivity, low rate capability and voltage decrease [85]. PANI provides novel feasibility for designing electrodes of rechargeable batteries due to its high flexibility, high electrical conductivity and low-cost synthesis. Moreover, lots of research has proved the applications of PANI for improving the electrochemical performance of rechargeable batteries, thus PANI is widely used in that field. In this chapter, we will emphatically discuss the three types of rechargeable batteries that are most applied and studied: lithium-ion batteries (LIBs), lithium-sulfur batteries (LSBs) and sodium-ion batteries (SIBs). The electrode materials’ design for LIBs, LSBs and SIBs that are associated with PANI will be also reviewed in detail.

### 3.1. Lithium-Ion Batteries (LIBs)

Among rechargeable batteries, LIBs are the most promising superstar used in portable electronic devices due to their high power/energy density, good portability, excellent cycling stability and environmental friendliness. As a rising secondary batteries, LIBs mainly rely on lithium ions moving between the positive electrode (cathode) and the negative electrode (anode) to operate. When it is charging, Li^+^ ions transfer from the cathode to the anode through the electrolyte, then results in a Li-rich state in an anode; when discharging, it is the contrary, that is, the cathode is in a Li-rich state. The mechanism is illustrated in Figure 12.

Since the cathode and anode provide the space for Li^+^ ions transfer, many researchers have been trying to seek electrode materials with superior electrochemical performance to optimize the characteristics of LIBs. In the next section, we will discuss the recent research progress on PANI modified cathode/anode materials for LIBs in detail.

#### 3.1.1. PANI Modified Cathode Materials

As one of the most promising energy storage devices, LIBs have been extensively investigated and rapidly developed since the 1970s [86]. In general, most cathode materials for LIBs are made from Li-containing transition metal oxides, furthermore, the higher Li content in the compound endows the cathode with better electrochemical performance, whereas Li-rich TMOs are the promising cathode materials. However, relatively low conductivity, cycling instability and structural instability are challenges that remain to be tackled. Fortunately, PANI with high conductivity, good stability and excellent flexibility can overcome these problems, so PANI modified cathode materials are wanted. Up to date now, Li-rich cathode materials have undergone various generations like LiM_x_O_y_ (M = Co, Ni, Mn, V), LiFePO_4_, Li(Ni_1−x_Co_x_)O_2_, Li(Ni_1−x−y_Mn_x_Fe_y_)O_2_ and Li(Ni_1−x−y_Co_x_M_y_)O_2_ (M = Fe, Al, Mn). Among them, LiCoO_2_, LiFePO_4_, LiV_3_O_8_, Li(Ni_x_Co_y_Mn_1−x−y_)O_2_ and Li(Ni_x_Mn_y_Fe_1−x−y_)O_2_ have been commonly chosen to prepare PANI modified cathode materials for LIBs.

As the first generation of the promising cathode materials for LIBs, LiCoO_2_ has been commercially applied due to its high Li content and stable cyclic behavior. In fact, LiCoO_2_ is dispersed onto the electrode surface with a solid powder, it is necessary to dope conductive medium into the electrical wiring of LiCoO_2_ powder [87]. PANI is suitable to act as the conductive medium due to its high electrical conductivity. Karima et al. [87,88] developed a Pickering emulsion route to produce PANI/LiCoO_2_ nanocomposites with well-ordered layered structure. Conductive and flexible PANI additive can connect the LiCoO_2_ powders well, and shorten charge transport pathways, thereby realizing increased conductivity and capability. As a consequence, all of the nanocomposites with various mass ratio of PANI showed enhanced specific capabilities compared with that of pristine LiCoO_2_ of 136 mAh/g.

LiFePO_4_ (LFP) is the next generation of promising cathode material after LiCoO_2_ owning to its excellent cycling stability, low cost, security and large capacity. However, blaming for its 1D channels for Li extraction [89], there are still two limitations including poor electronic conductivity and slow Li^+^ ion diffusion that hinder its applications for LIBs [90]. Carbon coating is a useful strategy to improve the electronic conductivity of LFP, but a C-LFP composite is electrochemically inactive, fortunately, electrochemically active PANI is promising to form a PANI/C-LFP composite by combining with C-LFP or substitute C-LFP for PANI-LFP composite. In Su’s work [91], a PANI-CSA (camphorsulfonic acid)/C-LFP composite was prepared by coating C-LFP with PANI-CAS in m-cresol solution. The composite cathodes delivered enhanced specific discharge specific capacity and rate capability. In particular, 10% PANI-CSA/C-LFP achieved a specific capacity value as high as 165.3 mAh/g. They attributed it to the electrochemically active PANI additives. Recently, Fagundes and coworkers [92] reported a PANI/LFP composite cathode material for LIBs. The composite was fabricated via an alternative synthesis route, which was beneficial to enhanced electrochemical properties and fast electron transfer rate of the LPF. Therefore, the composite showed a lower oxidation and reducing potential (Δ*E*_p_ = 0.20 V) than that of LFP (Δ*E*_p_ = 0.41 V) according to the CV. Gong et al. [93] adopted a PANI/poly (ethylene glycol) (PEG) co-polymer to simultaneously modify the electronic conductivity and Li ion diffusion rate of the LFP. The modification was derived from the synergistic effect between PANI and PEG: PANI served as a superior conductive medium, while PEG served as an excellent solvent for lithium salts and became the best known polymer ionic conductor. In return, the PANI/PEG co-polymer modified C-LFP based cathode achieved a high specific capacity of 125.3 mAh/g at 5 °C, as well as excellent cycling stability of 95.7% capacity retention after 100 cycles at 0.1 °C.

LiV_3_O_8_ is well-known as a promising cathode material for LIBs due to its large capacity, being chemically stable and its low cost, but pure LiV_3_O_8_ suffers from a short cyclic life and poor rate capability. Coating PANI on LiV_3_O_8_ is of great use to make up for the disadvantages of pure LiV_3_O_8_. Guo et al. [94] chemically synthesized a LiV_3_O_8_/PANI nanocomposite via the oxidative polymerization method. In the nanocomposite cathode, PANI coating acted as a conductive network structure, moreover well-crystallized regions and amorphous-like regions coexist in it, both of which boost electron transfer and lithium ion chemical coefficients, then resulting in better electrochemical properties than pristine LiV_3_O_8_, which included an enhanced cyclic stability (about 95% capacity retention after 55 cycles) and superior rate capability.

In recent years, the trimetallic Li-containing oxides cathode materials like Li(Ni_x_Co_y_Mn_1−x−y_)O_2_ and Li(Ni_x_Mn_y_Fe_1−x−y_)O_2_ arouse lots of interest because of their well rate capability, high specific capacity, low cost and environmental friendliness. However, with the increase of Ni content, they usually undergo the evaporation loss during the Li ion calcination and inferior cyclic stability due to residual Li_2_CO_3_ and LiOH impurities derived from side reactions after every cycle [95]. Likewise, coating PANI onto monometallic and bimetallic Li-containing oxides to form modified cathode materials can overcome these obstacles. Song et al. [96] obtained PANI modified PANI-coated Li(Ni_0.8_Co_0.1_Mn_0.1_)O_2_ cathode material for LIBs via a solution method (Figure 13c) and evaluated its electrochemical performance. PANI coating can remove the residual Li compounds like Li_2_CO_3_ and LiOH (Figure 13a,b) effectively and optimize the interfacial electrochemical reactions. Thus, the PANI modified cathode achieved a high initial discharge specific capacity of 193.8 mAh/g with a well capacity retention of 96.25% after 80 cycles at 1 °C and remarkable rate capability at high current density (1 °C, 2 °C and 5 °C). Karthikeyan and his coworkers [97] tested the hybrid PANI/Li(Ni_1/3_Mn_1/3_Fe_1/3_)O_2_ (0.2 mol PANI) cathode’s electrochemical characteristics in half-cell configuration. As a consequence, it maintained 86% of initial discharge capacity after 40 cycles, particularly, it can exhibit remarkable cyclic ability at ultra-high current densities of 5, 30 and 40 °C. They attributed it to the PANI additive, that is, PANI highly enhanced the conductive nature of the half-cell system and enabled efficient insertion and extraction of the Li ion.

In addition to PANI modified cathode materials, other PANI based composites like PANI/Zn [98], PANI/Cu/Li_2_MnSiO_4_ [99] and PANI@RGO/PW12 [100] have been recently used as cathode materials for LIBs due to their unique electrochemical characteristics. Furthermore, they achieved enhanced electrochemical properties including specific capacity, cyclic stability and rate capability, hence they are significantly hot directions for cathode materials of LIBs.

In fact, PANI itself can also be employed as a cathode material for LIBs. Based on high conductivity and flexibility provided by PANI, together with short charge transport pathways provided by the nanostructure, the PANI nanowire arrays as cathode material for LIBs delivered an initial discharge capacity of 159.83 mAh/g and exceptional cycling stability (119.79 mAh/g retained after 100 cycles at 30 mA/g) [101].

#### 3.1.2. PANI Modified Anode Materials

To date, common anode materials for LIBs include silicon based materials (Si and SiO_x_), metal oxides (SnO_2_, TiO_2_, NiO, Fe_2_O_3_ and Fe_3_O_4_) and metal sulfides (MoS_2_ and SnS_2_). Nevertheless, they are usually related to poor conductivity, structural instability and ease of self-aggregation. As like cathode materials, these obstacles can be also overcome by the PANI coating.

Silicon (Si) anode material for LIBs has been a promising research focus because of its highest ever-known theoretical capacity of 4200 mAh/g. However, Si is a semiconductor of low conductivity, additionally, it usually undergoes severe volume expansion (>300%) during intercalation and extraction of the Li ion [102]. Promisingly, the two problems can be greatly solved by PANI coating. Many efforts [102,103,104,105,106,107] have been devoted to modify the Si anode by fabricating a Si/PANI core–shell nanocomposite anode. In the Si/PANI core–shell structure, Si and PANI act as core and shell respectively, and PANI is tightly anchored to nano-Si by a covalent bond between PANI and Si. A PANI-encapsulated shell provides a large space for volume expansion and shrinkage of Si core during intercalation and extraction of the Li ion, which promotes the contact of electrode materials, then brings forth higher reversible capacity and better cycling stability. Besides, the coating of the conductive PANI layer significantly enables a fast Li ion and electronic transfer, thus resulting in excellent conductivity. In addition, doping PANI/Si composites with other active materials is another smart strategy to modify Si anodes’ electrochemical properties. Carbon materials (graphite [108], graphene [109,110,111], RGO [112] and CNTs [113]) and metal oxides (CeO_2_ [114] and TiO_2_ [115]) are conventionally chosen to the active materials. Graphite with powders facilitates better disperse of Si nanoparticles and electric contact of active materials, which effectively improves electrochemical properties, so the ternary PANI/Si/graphite composite can achieve a high initial capacity of 1392 mAh/g and stable capacity retention of 62.2% after 95 cycles [108]. Graphene and RGO sheets can be tightly attached to the PANI layer through intimate enhancement of π conjugation between them. The layer–layer structure of PANI/graphene (or RGO) sheet can encapsulate Si nanoparticles to form a new core–shell architecture [109,111] or glue nano-Si to form sandwich-like nanoarchitecture [110,112] with high elastic modulus and high tensile strength. Both of the architectures provided conductive and protective 3D network to avoid structural damage derived from volume expansion during intercalation and extraction of the Li ion. Additionally, the 3D network can promisingly promote electron and Li ion transfer, and then result in enhanced conductivity. A unique CNTs/PANI/Si composite with core–sheath architecture based on CNTs foam exhibits enhanced capacity and stability in Zhou’s study [113]. The 3D interconnected porous CNTs provide substantial conductive channels for the Li ion and electron transfer. Moreover, a PANI sheath offered Si with enough space to expand, ensuring structural integrity of electrode during Li ion insertion and extraction. In return, an enhanced initial specific capacity of 1954 mAh/g and stable cyclic ability (727 mAh/g maintained after 100 cycles) were achieved. There are similar synergistic effects that exist in the Si@PANI@CeO2 composite [114] and Si@PANI@TiO2 composite with a double core–shell [115] structure: on the one hand, PANI served as a protecting electrode against structure damage derived from volume expansion of Si, on the other hand, PANI and lithiated TiO_2_ or CeO_2_ highly promoted electronic transport. As an outcome, modified anodes exhibited enhanced conductivity and cycling stability compared to pristine Si and PANI/Si composite.

Silicon oxide (SiO_x_) is a type of rising anode material for LIBs in recent years, but like Si, low conductivity and large volume expansion during lithiation/de-lithiation severely hinder its practical applications. These obstacles can be well tackled by designing core–shell sandwich-like PANI-wrapped SiO_x_/CNTs [116] or fabricating SiO_x_/PANI/Cu_2_O composite anodes [117,118]. In the dual core–shell sandwich structure PANI/SiO_x_/CNTs composite (as seen in Figure 14) [116], the CNTs enhance the conductivity of SiO_x_ and the PANI endures large volume expansion during lithiation/de-lithiation of SiO_x_. Owing to the synergistic effect between PANI and SiO_x_, a high charge/discharge capacity of 1156/1178 mAh/g at 0.2 A/g and 728/725 mAh/g retained after 60 cycles at 8 A/g was achieved. A hollow SiO_x_ coated with PANI/Cu_2_O nanocomposite can be prepared by the Stober method [117,118]. The nanostructure effectively relieved the volume expansion during lithiation/de-lithiation, moreover, PANI and Cu_2_O dual-coating significantly enhanced reversibility and conductivity of SiO_x_ and prevented it dropping from the electrode surface.

SnO_2_ has been actively employed as a promising anode material for LIBs due to its high theoretical specific capacity of 790 mAh/g, low discharge potential, low cost and natural abundance. However, there are still three problems that hamper its commercialization: (1) poor cycling stability; (2) low electronic and ionic conductivity and (3) enormous volume expansion (>200%) during lithiation/de-lithiation process. Intensive efforts have been devoted to tackle these disadvantages by synthesizing PANI/SnO_2_ based ternary composites. Guo et al. [119] reported an in-situ polymerization sol–gel route to prepare SnO_2_-Fe_2_O_3_@PANI composite. The growth of SnO_2_-Fe_2_O_3_ particles was firstly suppressed by the PANI on their outer surface during polymerization, next the full coating of a carbon shell encapsulated the Fe_2_O_3_ particles in the thermal treatment, which forms a unique SnO_2_-Fe_2_O_3_@C structure, in which SnO_2_-Fe_2_O_3_ particles were tightly coated with PANI and the outer PANI shell effectively restricts their agglomeration, resulting in enhanced stability. Additionally, the introduction of a carbon layer achieves improved electronic conductivity. Hence the unique structure of the SnO_2_-Fe_2_O_3_@C nanocomposite significantly improves its electrochemical properties, achieving the fully reversible reaction and alloy reaction of SnO_2_. Enhanced capacity retention of over 1000 mAh/g at 400 mA/g after 380 cycles and excellent rate performance of 611 mAh/g at 1600 mA/g were reported. A novel 3D ternary PANI/SnO_2_/RGO nanostructure was successfully designed as an anode for LIBs via an easy dip-coating of PANI@SnO_2_ and graphene dispersion on Cu foam (Figure 15c) in Ding’s [120] study. In the nanostructure, PANI acted as the conductive matrix as well as the glue that bind the hollow SnO_2_ nanoparticles on RGO sheets tightly to avoid aggregation while cycling, which greatly improved the rate performance; the hollow SnO_2_ nanoparticles acted as the buffer for enormous volume changes during insertion/extraction of Li, and provided active spots for vitiation, which resulted in enhanced cycling stability; the assembly of PANI@SnO_2_, RGO and Cu foam with strong contact achieves ultra-fast electron transport by a 3D expressway, which effectively enhances electronic conductivity and rate performance. As predicted, the nanocomposite exhibited excellent rate performance of 268 mAh/g at 1000 mA/g and cycling stability (749 mAh/g of initial 772 mAh/g was retained after 100 cycles at 100 mA/g), much higher than SnO_2_/RGO, PANI/SnO_2_ and pure SnO_2_ (Figure 15a,b). Yi et al. [121] synthesized 3D expanded graphite (EG)/PANI /SnO_2_ composite by the solvothermal method followed by in-situ oxidative polymerization. The long-ordered 3DEG layer structure greatly endured volume expansion of SnO_2_, PANI can reduce the electric contact between electrode materials, the double synergies brought forth enhanced electrochemical characteristics including a high initial reversible capacity of 1021 mAh/g at 0.1 A/g and 408 mAh/g retained after 100 cycles. Recently, Wang and coworkers [122] designed a novel 1D PANI@SnO_2_@MWCNT composite as the anode material for LIBs. The MWCNT can provide convenient electron transfer channels and an effective buffer for huge volume changes of SnO_2_, furthermore, the synergy of the conductive PANI and MWCNT skeleton significantly enhanced the electronic and ionic conductivity of the ternary 1D PANI@SnO_2_@MWCNT composite. In return, the composite delivered a high reversible charge/discharge specific capacity of 878/888 mAh/g at 0.2 A/g over 100 cycles and 524/527 mAh/g at 1.0 A/g over 150 cycles.

Compared to silicon based materials and SnO_2_, TiO_2_ is a superior candidate for Li anode materials as a result of its small volume expansion (<4%), reliable security and excellent cycling stability. Unfortunately, the nano-TiO_2_ tends to agglomerate and decompose while cycling, which result in capacity fading and decreases in electroactive sites. As a consequence, the rate of lithiation/delithiation slows down. Doping the conductive PANI/RGO [123] or PANI/GO [124] phase in TiO_2_ is an effective way to resolve it. Both of PANI/RGO and PANI/GO can be constructed into a sandwich structure with TiO_2_. In the TiO_2_/PANI/RGO sandwich structure, the TiO_2_ nanoparticles sandwiched between PANI and RGO nanosheets, which could effectively prevent the agglomeration of TiO_2_ nanoparticles and enable fast insertion and extraction of the Li ion. As a result, the composite anode showed enhanced rate performance and cycling stability (discharge capacity of 149.8 mAh/g accompanying Coulombic efficiency of 99.19% at 1000 mA/g after 100 cycles) compared with pure TiO_2_ [123]. Different from the former, PANI nanorods were vertically grown on both sides of amorphous TiO_2_-GO nanosheets to form a stable TiO_2_/PANI/GO sandwich structure. The GO network provides many conductive channels for electron transport and allowed nanoscale PANI and TiO_2_ to settle well onto the GO network to form a stable sandwich structure, which led to enhanced electronic conductivity and stability. An excellent initial discharge capacity of 1335 mAh/g at 50 mA/g and 435 mAh/g after 250 cycles at 100 mA/g were reported [124].

NiO is known as a semiconductor of low conductivity, and the nanoscale NiO particles are easily converted to inactive Li_2_O through the discharge reaction: NiO + 2Li = Ni + Li_2_O, leading to poor electric contact between Li-active nanoparticles and substrate. Moreover, NiO nanoparticles tend to agglomerate and form inactive bulk particles while cycling. In order to tackle it, Huang et al. [125] developed a nickel foam-supported NiO/PANI composite as an anode material for LIBs. It was with enhanced conductivity and stability by gluing NiO flakes tightly on the PANI layer, which can prevent NiO loss caused by a side effect. Furthermore, weaker polarization ensures better reversibility and cycling performance. The NiO/PANI electrode retained 520 mAh/g after 50 cycles at 1 °C, higher than pristine NiO electrode of 440 mAh/g. The NiO/PANI composite core–shell arrays [126] could work better than the former. It not only exhibited weaker polarization, but also showed excellent Li ion storage capability and enhanced electronic conductivity were derived from the introduction of conductive PANI network that could enable fast electron and ion transfer and improve structural stability. Therefore, an enhanced specific capacity of 780 mAh/g at 0.1 A/g, superior rate capability and cycling stability to bare NiO were observed (Figure 16).

As a kind of metal oxide of natural abundance, high theoretical capacity (1007 mAh/g) and low cost, Fe_2_O_3_ has been popularly developed as anode materials for LIBs. Ma et al. [127] successfully synthesized PANI-coated hollow Fe_2_O_3_ nanoellipsoids via a solvothermal technique followed by a post-coating process. The porous and hollow structure can effectively accommodate the volume change of Fe_2_O_3_, while PANI coating acted as a conductive medium that greatly improved the electronic and ionic conductivity in addition to buffering the expansion/contraction of the active material during electrochemical reactions. Consequently, the PANI/Fe_2_O_3_ composite delivered charge capacities of 366, 223.4 and 105.8 mAh/g at 0.5, 1.0 and 2.0 C respectively, and maintained a charge capacity of 412.1 mAh/g after 150 cycles at 0.2 °C, indicating enhanced rate capability as well as extended cyclic lifespan.

Fe_3_O_4_, namely magnetite, is newly developed as the PANI modified anode material for LIBs. Similar to Fe_2_O_3_, natural abundance, high theoretical capacity (926 mAh/g) and low cost are its attractions, but huge volume expansion and severe particle agglomeration while cycling need to be overcome. A unique Fe_3_O_4_@PANI composite with yolk–shell micro-nanoarchitecture was obtained by Wang’s group [128]. Figure 17 vividly illustrates its formation process. The micro/nanostructure is favored for preventing the bulk Fe_3_O_4_ from aggregation while cycling, the porous yolks and void spaces can shorten transport length for Li ions and electrons, and also provide extra sites for ion storage, and the PANI layer can effectively improve the conductivity, resulting in enhanced electrochemical performance. As expected, a high reversible capacity of 982 mAh/g after 50 cycles at 100 mA/g and an outstanding rate capability of 734.6 mAh/g at 1000 mA/g were achieved. Coating layered graphene onto PANI nanofiber-anchored Fe_3_O_4_ is also a smart strategy [129]. It was demonstrated that the graphene/Fe_3_O_4_/PANI showed a superior reversible specific capacity of 1214 mAh/g, extraordinary rate capability, low volume expansion, enhanced cycling stability and 99.6% coulombic efficiency over 250 cycles, owing to the collective effect of layered graphene, Fe_3_O_4_ hollow rods, as well as the superior conductivity of PANI.

MoS_2_ possesses a unique 2D layered structure that enables fast Li insertion and extraction. However, MoS_2_ still undergoes poor rate performance and fast capacity decay because of poor electrical conductivity between S–Mo–S sheets while used as an electrode material. An effective improvement is to hybrid MoS_2_ with conductive additives like conducting PANI [130] that can greatly improve the conductivity and stability. The 3D hierarchical MoS_2_/PANI nanoflowers (Figure 18) were prepared by a simple hydrothermal method [131]. Such hierarchical architectures provided sufficient void space for the Li ion to diffuse and ensured structural integrity of the electrode material. The flexible PANI chains not only improved electrical conductivity, but also maintained hierarchical architectures of the nanoflowers during heat treatment. Enhanced electrochemical performance was achieved due to the synergy of MoS_2_ and PANI, as well as the unique 3D hierarchical structures of MoS_2_/PANI nanoflowers.

SnS_2_ has a similar layered structure like MoS_2_ that is beneficial to Li ion insertion. The 2D SnS_2_@PANI nanoplates with a lamellar sandwich nanostructure can provide a good conductive network between neighboring nanoplates, shorten the path for ion transport in the active material and alleviate the expansion and contraction of the electrode material during charge–discharge processes, resulting in enhanced electrochemical performance. As a consequence, the SnS_2_@PANI nanoplate electrode delivered a high initial reversible capacity of 968.7 mAh/g, excellent cycling stability (75.4% capacity retention after 80 cycles) and an outstanding rate capability (356.1 mAh/g at 5 A/g) [132].

In recent years, PANI modified binary metal or binary metal oxides composites have been tried to employ as anode materials for LIBs. Such as PANI/Sn-Cu nanotubes [133], PANI/Co_3_O_4_-CuO [134] and PANI/Cu_3_Mo_2_O_9_ [135], in these architectures, the PANI layer effectively relieves the stress associated with volume changes of the binary compounds and improves conductivity. Furthermore, unique composite structures help a lot. For example, 3D porous PANI hydrogel/Sn-Cu nanotubes structure provides network for electron and Li ion transport, resulting in improved electrical conductivity; the PANI/Co_3_O_4_-CuO raspberry design can result in lots of advantages like suppressed agglomeration, an effective electrical contact, enhanced cycling stability, as well as a lower charge transfer resistance while cycling. As a result of the synergistic effect, enhanced electrical performance including reversible capacity, cycling lifespan and rate capability were achieved [134].

### 3.2. Lithium-Sulfur Batteries (LSBs)

Lithium-sulfur batteries (LSBs) are the next generation of promising rechargeable batteries with high specific energy of 2600 Wh/kg and high theoretical specific capacity of 1675 mAh/g. Conventional LSBs choose metal Li as the anode material and sulfur as the cathode material, sulfur can react with Li to form Li_2_S. However, the process of reduction reaction is very complicated, many side reactions are involved in the reduction, as a result, many residual side products from S_8_ to soluble lithium polysulfides Li_2_S_x_ will be reduced on the Li anode in a parasitic reaction, resulting in the shuttle mechanism and low coulombic efficiency [136]. Furthermore, the conductivity is very low due to the insulating nature of sulfur, and the sulfur cathode usually suffers from huge volume change while cycling, leading to poor cycling stability and inferior rate performance. Considerable efforts have recently been made to modify the electrochemical performance of the sulfur cathode like adding an absorbing agent to absorb polysulfides, designing Li-protecting separators and encapsulating sulfur in conducting polymer matrix. PANI coating is suitable to encapsulate sulfur in the PANI matrix owning to its flexibility, high conductivity, slight solubility in organic electrolyte and porous architecture.

The process of encapsulating sulfur in PANI matrix to form PANI/S composite can be called the vulcanization reaction. During the reaction, partial S atoms substitute H atoms on the aromatic rings by reacting with the unsaturated bonds in PANI chains during heat treatment, then inter/intra-chain disulfide bonds are formed on the side chain. Yan et al. [137] designed a nano-porous sulfur/PANI (SPANI) composite by in situ chlorinated substitution and vulcanization reactions. In the main backbone chain in SPANI, sulfur was efficiently encapsulated in the PANI chain, and the obtained SPANI chain provided electronic conductivity and electrostatic attraction force to stabilize polysulfide anions while cycling, moreover, the disulfide side chain can act as the second electrochemical redox component, resulting in enhanced capacity behavior. Consequently, excellent reversible capacity of 750 mAh/g, superior cycling stability (89.7% capacity retention after 200 cycles at 0.3 °C) and high rate performance were observed. Duan and coworkers [138] produced a PANI-coated sulfur composite cathode for LSBs by the layer-by-layer assembly method, followed by the crosslinking and heat treatment process. The outer PANI shell can protect the sulfur (S_8_) and polysulfides from dissolution into Li anode, while allowing Li to be permeable during the charge/discharge process. Furthermore, the conductive PANI layer provides substantial conducting channels for electron transport. The final product showed a high conductivity of 0.23 S/cm.

Structural instability while cycling derived from sulfur severely lower the cycling stability, additionally, soluble lithium polysulfides have been a potential hazard to rate capability. Hence it is crucial to design a rational structure to surpass these disadvantages. Ma et al. [139] prepared a hollow PANI sphere/sulfur composite via a vapor phase infusion technique. The sulfur was deposited onto both of the inner and outer of the PANI hollow sphere, the void space in the PANI/sulfur nanoparticles effectively buffered volume expansion while cycling (Figure 19). The PANI shell also prevented the dissolution and migration of polysulfides, as well as improved the electronic and ionic conductivity. Moreover, the chemical bond between PANI and S was formed during heat treatment, which inhibited the shuttle effect. As predicted, excellent overall electrochemical performance including high reversible capacity of 602 mAh/g after 1000 cycles at 0.5 °C and coulombic efficiency as high as 97% were realized. Zhou et al. [140] obtained PANI/S yolk–shell structure nanocomposite by in situ polymerization, followed by the heat treatment process. In the yolk–shell structure, sulfur as the yolk was encapsulated in PANI shell, which helped to immobilize the polysulfides and accommodate the volume expansion of sulfur, as a result, excellent performance including cycling stability and high capacity retention were obtained.

Encapsulating sulfur/carbon composites in PANI coatings is another effective way. Different from vulcanized PANI/S composites as the former, the in situ polymerization of PANI coatings on the sulfur/carbon composites can be accomplished without heat reaction. Derived from promising advantages of both PANI and carbon materials, the ternary S-C@PANI hybrid cathodes for LSBs hold promise for superior electrochemical performance due to a superior synergistic effect. PANI can be polymerized in situ onto the graphene (G) and GO sheet to obtain the PANI-G and PANI-GO membrane, the sulfur nanoparticles are sandwiched between the membranes to form a ternary sandwich structure. The conductive PANI-G and PANI-GO networks not only buffered a huge volume expansion of the sulfur, but also mitigated the diffusion of lithium polysulfides to the Li anode by a chemical interaction between the imine group (–N=) of the quinoid ring and polysulfides, finally resulted in excellent electrochemical performance [141,142]. CNTs are widely employed as supporting materials because of high conductivity, large specific surface area and excellent mechanical properties. Considering the advantages of PANI and CNTs, it is a smart strategy to develop a ternary composite where sulfur supported by MWCNTs and coated with PANI, that is, the MWCNTs-S@PANI as a cathode material for LSBs. The unique sandwich architecture effectively avoided dissolution and diffusion of polysulfides, and the MWCNTs provided conductive network, flexible PANI accommodated volume change while cycling, resulting in enhanced cycling behavior and rate capability [143]. Porous carbon materials like activated carbon and mesoporous carbon can provide an efficient conductive network for S, while PANI coating further reduces the volumetric effect of S and facilitates electronic conduction, as well as prevents lithium polysulfides from dissolving in an electrolyte, thus the composite electrodes exhibited enhanced electrochemical characteristics [144,145]. In addition, acetylene black is also a promising carbon material to form PANI@S-acetylene black composite as cathode for LSBs [146]. S-acetylene black powder was encapsulated in PANI coating as the shell, which accommodated volume expansion while cycling. Moreover, the efficient conductive network provided by the acetylene black, together with the strong affinity to sulfur and polysulfides provided by PANI, enabled the uniform dispersion of the sulfur, promoted the transportation of ions and enhanced the cyclic performance of the LSBs.

### 3.3. Sodium-Ion Batteries (SIBs)

Similar to LIBs, sodium-ion batteries (SIBs) rely on the insertion and extraction of Na ion to operate. Compared to Li element, Na element is of higher natural abundance and lower cost, hence SIBs have gained lots of recognition as an effective alternative to LIBs. As well known to us, electrochemical performance of rechargeable batteries is largely determined by electrochemical properties of electrode materials. Thus the SIBs desirably require us to be equipped with advanced electrodes. Just like the other secondary cells, PANI holds much promise for SIBs due to its versatile electrochemical properties. Similarly, in the next section, we will overview the recent development on PANI modified cathode/anode materials for SIBs in detail.

#### 3.3.1. PANI Modified Cathode Materials

Iron phosphate (FePO_4_) is an important cathode material investigated for SIBs. However, poor electronic conductivity and inferior ionic diffusion severely hinder its commercialization. Carbonized PANI nanorods (CPNRs) are of great potential for enhancing electrochemical properties of FePO_4_ [147]. The CPNRs/FePO_4_ composite was synthesized by a carbonization process of the PANI/FePO_4_ composite, which was prepared through a microemulsion method. In the as prepared CPNRs/FePO_4_ composite, CPNRs provided high-speed pathways for electron transport, enabling fast charge transfer during the process sodiation/desodiation, moreover, FePO_4_ can load on the surface of CPNRs by a noncovalent bond, decreasing the resistance of charge transfer. As the cathode material, a high initial discharge specific capacity of 140.2 mAh/g, with the value being retained at 134.4 mAh/g after 120 cycles was achieved.

Na_3_V_2_(PO_4_)_3_ (NVP) is a promising cathode material for SIBs with a superionic conductor framework, large channels for charge transfer and excellent thermal ability. Nevertheless, bare NVP is poor of electrical conductivity, which greatly restricts its practical application. Chen et al. [148] designed double coating NVP (NVP@C@HC) to improve the electrochemical properties. Wherein, the HC was derived from pyrolytic PANI/NVP was tightly anchored to the surface of C and PANI coating via chelating interactions of citric acid and vanadate, which effectively suppresses agglomeration of NVP particles while cycling and enhances conductivity. Furthermore, the double carbon layers buffer volume change during the charge/discharge process. As expected, the double carbon-coated porous NVP@C@HC composite delivered an excellent rate capability (60.4 mAh/g at 50 °C) and a long-term cyclability (capacity retention of 83.3% at 40 °C after 3000 cycles).

Very recently, a novel Na(Ni_1/3_Mn_1/3_Fe_1/3_)O_2_ (NNMF) embedded on the conductive PANI backbone was demonstrated to exhibit enhanced electrochemical performance while used as a cathode for SIBs [149]. The uniform dispersion of NNMF on PANI ensured better electrical contact between electrolyte and active materials, moreover, short Na ion transfer pathways and buffered mechanical stress provided by the porous PANI network significantly lead to enhanced capacity behavior and cyclability.

Similar to LIBs, as a conductive polymer, PANI itself is also an ideal cathode material for SIBs. Nanostructured PANI like nanofiber PANI presented considerable specific surface area and electrical conductivity, moreover, PANI with high flexibility effectively improved structural stability and cycling stability [150,151].

#### 3.3.2. PANI Modified Anode Materials

Since Na ion (0.102 nm) is bigger than the Li ion (0.076 nm), it is more challenging to seek a suitable host anode material for SIBs. In order to cope with this challenge, modified PANI based materials with advanced electrochemical performance have recently been employed as anode materials for SIBs.

SnO_2_ anode performance has been demonstrated to achieve long cyclic life and excellent rate performance by forming a core–shell structured SnO_2_ hollow sphere (SnO_2_-HS)/PANI composite electrode [152]. The unique hollow structure of the SnO_2_ core and the flexible PANI buffer layer can alleviate volume expansion of SnO_2_ and aggregation of generated Sn particles during cycling. Therefore, a high reversible capacity of 213.5 mAh/g over 400 cycles at 300 mA/g was delivered.

Transition metal sulfides like Co_3_S_4_ and MoS_2_ have been used to fabricate the PANI modified anode for SIBs. Co_3_S_4_@PANI nanotubes were formed via the uniform coating on both outer and inner surfaces of Co_3_S_4_ nanotubes, which were obtained by a facile self-template hydrothermal route based on the Kirkendal effect [153]. The conductive PANI layer enables electron and Na ion transport and prevents Co_3_S_4_ nanotubes from structural collapse or pulverization while cycling. Finally, the composite achieved a high maintained capacity of 252.2 mAh/g of after 100 cycles at 200 mA/g, much higher than that of bare Co_3_S_4_ nanotubes (58.2 mAh/g retained of initial value of 815.3 mAh/g after 100 cycles at 200 mA/g). The interlayer spacing of 2D MoS_2_ can be enlarged by the introduction of the conductive PANI layer to form a MoS_2_/PANI heterostructure [154]. The enlarged interlayer spacing remarkably facilitates large Na ion diffusion, as well as improves conductivity. Furthermore, the inter-overlapped MoS_2_/PANI nanosheets can retain stable structural integrity while cycling due to the strong coordination ability between Mo and nitrogen atoms. As a consequence, the hybrid anode exhibited high capacity, rate capability and long cyclic life.

In addition to directly functioning as active anode materials for SIBs, carbonized PANI is also extensively utilized due to superior Na-storage performance. For example, PANI was also investigated as the SIB anode in a PANI carbonized 3D porous carbon-coated graphene hybrid system [155], where SiO_2_ and PANI layer were successively deposited on the surface of 3D porous graphene (3D PG; Figure 20A). The unique architecture that is composed of 3D PG networks and a porous PANI-converted carbon coating can endow the hybrid with high electrical conductivity, rapid ion intercalation, substantial active sites, short ionic diffusion pathways and high structural stability for efficient Na-storage. Consequently, the 3D PG@C composite displayed remarkable Na-storage performance, initial discharge capacity as high as 824 mAh/g at 50 mA/g, high reversible capacity, enhanced cycling stability (323 mAh/g after 1000 cycles at 1000 mA/g) compared to pure 3D PG and carbon (Figure 20B), along with excellent rate capability (207 mAh/g at 10 A/g).

As a N-rich CPs, S and N co-doped S/N carbon nanotubes (S/N-CT) system can be formed via carbonization of an S-containing PANI derivative [156]. N in carbonized PANI enables efficient Na adsorption performance and high electrical conductivity to enhance Na-storage performance; the introduction of S into a carbon matrix can further enlarge interlayer spacing, offer active sites and shorten ion diffusion distance to improve rate performance and cycling stability. In agreement with the prediction, the as-prepared S/N-CT anode for SIBs delivered a reversible capacity as high as 340 mAh/g at 0.1 A/g and an excellent cycling stability (94% capacity retention after 3000 cycles at 5 A/g).

Table 2 presents the preparation method and electrochemical performance of some typical PANI modified rechargeable batteries electrode materials.

## 4. Applications of PANI for Fuel Cells

As a type of novel energy storage and conversion devices of green, sustainable and efficient energy storage, fuel cells have been actively developed during these years. Unlike supercapacitors and rechargeable batteries, the fuel cells can realize the direct conversion from chemical energy to electrical energy. Common fuel cells including direct methanol fuel cell (DMFC), proton exchange membrane fuel cell (PEMFC), polymer electrolyte membrane fuel cell (PEMFC), alkaline fuel cell (AFC), Zn air cell and microbial fuel cell (MFC) have attracted substantial research from scientists in the last decade. CPs hold a lot of promise for use as electrocatalysts of fuel cells due to their high conductivity, tunable morphologies and high flexibility. These CP supported and CP derived electrocatalysts usually show high catalytic activity and good stability. Among CPs, PANI based electrocatalysts have been demonstrated to deliver the excellent catalytic activity in a hydrogen oxidation reaction (HOR), oxygen reduction reaction (ORR), hydrogen evolution reaction (HER) and methanol oxidation reaction (MOR). In this section, we will give a summary of recent research of PANI supported metal electrocatalysts and PANI derived metal-free electrocatalysts.

### 4.1. PANI-Based Supported Metal Electrocatalysts

PANI with high conductivity, various tunable morphologies and high flexibility can effectively enhance the conductivity, catalytic activity and stability, so it has been actively used to support most metal electrocatalysts. Platinum (Pt) is the most used metal electrocatalysts as a result of extraordinary MOR, HER and ORR catalytic activity [157,158]. PANI with a nanowires network structure was used as support for Pt dispersion in order to enlarge the application of Pt [159]. The PANI nanowires network architecture was beneficial to distribute Pt nanoparticles and to create conductive channels, leading to form a hybrid nanocatalyst with excellent electrocatalytic activity in MOR. In addition to individual PANI supported Pt catalysts, PANI based composites like PANI/C, PANI/MWCNTs and PANI/SnO_2_ have been investigated as stable and MOR catalytic systems [160,161,162].

Nevertheless, high cost of Pt is the main problem that limits its commercial utilization, thus it is desirable to look for non-noble metals or their compounds as the alternatives to replace Pt catalyst. Yuan et al. [163] designed an efficient PANI/carbon black (PANI/C) composite-supported iron phthalocyanine (FePc) as an ORR electrocatalyst for FePc in an air-cathode single-chamber MFC. The resulting PANI/FePc/C composite catalyst exhibited better catalytic activity than bare Pt, PANI/C and FePc/C, reflecting PANI additive enhanced the activity of C in ORR. Moreover, the power per cost of the PANI/C/FePc catalyst was 7.5 times greater than that of the bare Pt catalyst, which indicated that the PANI/FePc/C composite can be a promising alternative to Pt catalyst in MFC. Zhang’s group [164] has demonstrated that a PANI supported MoS_2_ electrocatalyst achieved active HER catalytic performance. The flexible PANI effectively prevents MoS_2_ from aggregation, ensuring the uniform and vertical dispersion on the PANI branches with high edge exposure of MoS_2_ nanosheets. Apart from Mo and Fe, various non-noble metal electrocatalysts, such as Mn, Co, Ni and their compounds have been studied to substitute for Pt.

Proton exchange membrane fuel cell (PEMFC) is also a promising fuel cell of high power density. Pt-based electrocatalysts are common electrode materials for ORR and HOR. However, the main obstacle is still the dear cost of Pt. Many Pt-free non-noble metal-based catalysts have been discovered as novel catalysts with high ORR activity, but non-noble materials as HOR catalysts are rarely reported. Recently, Guo and coworkers [165] discovered that the Fe NPs-PANI/CNT catalyst synthesized by controllable self-assembly could be an appropriate HOR catalyst in the PEMFC. HOR kinetics can be written as H_2_→H_ad_→H^+^, the whole kinetic transfer efficiency is largely determined by the intermediate H_ad_. Promisingly, Fe-H_ad_ reversibility would make the Fe-based catalyst maintain stable in the acid condition. Furthermore, PANI/Fe NPs interface can effectively strengthen mass transfer and realized the recovery of active sites in the presence of conductive and flexible PANI support, driving HOR intermittently even at high potential. As a result, the H_ad_ adsorption/desorption process can be rapidly driven at low potential, resulting in the remarkable catalytic activity, power density as high as 161 W/kg and high durability. The work has paved the way for non-noble materials as HOR catalysts.

As mentioned above, PANI supported Pt shows good catalytic behavior in MOR, therefore the PANI/Pt catalysts could help a lot in direct methanol fuel cell (DMFC). For instance, Gharibi’s study [166] proved that the PANI/C (vulcan XC-72) supported Pt electrocatalyst exhibited better catalytic performance than traditional Pt/C with a nafion electrocatalyst in DMFC. In conventional DMFC, carbon XC-72 acts as support as Pt, while nafion acts as a binder and proton conductor. However, the agglomeration of carbon particles and slow charge transport severely hinder its practical application. Gharibi et al. substituted nafion for PANI nanowires. Conductive PANI nanowire structure acted as an efficient carrier for electron and proton transport, enhancing electrical conductivity and increasing methanol diffusion coefficient. Additionally, as a flexible binder, PANI with unique architecture significantly suppressed the agglomeration of carbon particles. The resultant Pt/C-PANI electrocatalyst showed better electrocatalytic performance than Pt/C with nafion electrocatalyst. Meantime, high cost of Pt might be a disadvantage, but it could be well overcome by the binary metal catalysts [160].

A novel poly(pyrrole-co aniline) (PPCA) hollow nanosphere (HN) supported Pd nanoflowers (Pd NFs) was designed for methanol electrooxidation [167]. The Pd NFs were electrodeposited from an aqueous solution of 0.01 M PdCl_2_ and 0.5 M H_2_SO_4_ in a facile one-step method, while the PPCA HN was obtained by in situ emulsion polymerization. Pd NFs on a PPCA HN coated glassy carbon electrode (GCE) was finally fabricated via the electrochemical method. Conductive PPCA HN co-polymer could surpassingly improve conductivity compared to exclusive conductive PANI or PPy, as well as enlarged specific surface area. The resulting Pd NFs/PPCA HN/GCE demonstrated better electrocatalytic activity than Pd NFs/PANI, Pd NFs/PPy and individual Pd NFs.

PANI-functionalized can be an effective support as metal electrocatalysts in fuel cells, especially in proton exchange membrane fuel cell (PEMFC). Pt/CNT catalysts show enhanced electrocatalytic activity and stability compared to Pt electrocatalysts for PEMFC. However, Pt nanoparticles were hard to deposit uniformly and directly onto highly graphitized CNT surface without active functional groups, the introduction of bridging conductive and stable PANI was capable of enhancing the binding strength between Pt and CNT by π–π bonding provided by PANI (Figure 21). Moreover, the Pt–N bonding endowed Pt nanoparticles with higher dispersion, whose size distribution ranged from 2 to 4 nm, bringing forth enhanced electrocatalytic activity of the resultant Pt-PANI/CNT catalyst [168].

A new Pt-C@PANI core–shell structured catalyst was developed for PEMFC. The thin PANI layer was directly polymerized onto the Pt-C surface. The unique The PANI-decorated core–shell architecture could induce electron delocalization between the Pt d orbitals and the PANI π-conjugated ligand accompanying with electron transfer from Pt to PANI, which explained the enhanced catalytic activity and durability. Furthermore, the Pt-C@PANI (30%) addressed the best catalytic activity and superior durability compared with the non-PANI-decorated Pt-C catalyst, indicating the thickness of PANI shell might have an influence on catalytic properties, in which the suitable thickness (5 nm) of the PANI shell greatly protected the carbon core from direct exposure to the corrosive surroundings [169].

### 4.2. PANI-Derived Carbon Based Metal-Free Electrocatalysts

As a N-containing carbon material, the properties of PANI are expected to get improved while doping heteroatoms (N, B, S, O and P) to the PANI chains, and PANI has been widely attempted to fabricate advanced metal-free catalyst support by doping heteroatoms to it. Heteroatom-doped PANI derived porous carbon-based catalysts are generally focused on modifying ORR activity since ORR is in huge demand for sustainable and non-noble element. PANI-derived N- and O-doped mesoporous carbon (PDMC) as a sustainable and non-noble ORR electrocatalyst has been demonstrated to deliver extraordinary electrochemical catalytic activity toward ORR [170]. PDMC was prepared by polymerizing PANI in situ within the pores of SBA-15 mesoporous silica, followed by subjecting PANI/SBA-15 to carbonization under an inert atmosphere, and finally etching away the silica framework (Figure 22). The final metal-free PDMC toward ORR showed even better electrocatalytic activity than Pt-PANI/SBA-15 at high current density and achieved preferable four electrons pathway toward ORR, which could be ascribed to the synergistic activities of N and O species that were implanted into it. It is suggested that the metal-free PDMC is promising to challenge conventional paradigms that Pt based catalysts. Inspired by Silva’s work, cheaper GO and graphene compared with SBA-15 were used to synthesize PANI-derived carbon-based PNCN and GNR/PANI metal-free electrocatalysts. Owing to high specific surface area and high N content of GO and graphene, as well as respective unique structures, ideal catalytic activities and stability were delivered for ORR [171,172].

In recent years, some researchers discovered that the electrocatalytic performance of the PANI derived metal-free catalysts could be further enhanced by co-doping of transition metals. In order to understand the mechanism on enhanced activation related to the transition metal dopants, Peng et al. [173] studied the effect of the addition of various transition metals (Mn, Fe, Co, Ni and Cu) on the structure and performance of the doped carbon catalysts M-PANI/C-Mela, accompanied with a metal-free catalyst as a reference. SEM showed that doping with Fe and Mn led to a graphene-like structure, while doping with Co, Ni and Cu led to a disordered or nanosheet structure. Catalysts doped with transition metals exhibited enhanced catalytic performance compared to the metal-free reference, moreover, their ORR activity followed the order of Ni < Mn < Cu < Co < Fe (Figure 23), which was consistent with the order of their active N contents. As suggested in the paper, the collective effect of the three aspects may result in the various performance enhancement of the transition metal dopants, that is, the N content/active N content, metal residue, as well as the surface area and pore structure of the catalyst. Nevertheless, the enhanced performance is not determined by any singer factor. We can further mean that the PANI would make an indispensable contribution due to the extensive active N sites provided by it.

## 5. Conclusions and Outlook

In this review, we presented here important research progress on the applications of PANI for electrochemical energy storage and conversion. Pure PANI with high conductivity, ease of synthesis, high flexibility, low cost, environmental friendliness and unique redox characteristics is a kind of active and economical electrode material for supercapacitors (pseudocapacitors) and rechargeable batteries (cathode material). Due to a considerable specific surface area, advantageous pore structures and high N content, PANI derived porous carbon materials are also suitable to serve as support for an electrocatalyst of fuel cells. However, while used as a supercapacitive electrode, poor cycling stability and inefficient capacitance contribution are delivered. Additionally, with the rapid development of energy, more stable molecular architecture, higher power/energy density and more N-active sites are greatly desirable, while individual PANI can not meet the ever-increasing demand. Therefore, it is necessary to combine PANI with other active materials like carbon materials, metal compounds and other CPs. In various PANI based composite structures, PANI generally acts as a conductive layer and network, and the resultant PANI based composites with various unique structures have exhibited superior electrochemical performance in supercapacitors, rechargeable batteries and fuel cells due to the synergistic effect. However, there still are some disadvantages that remain to be improved:PANI is hard to commercialize in the electrochemical field due to its relatively high cost and low practicability compared with conventional inorganic materials.PANI is hard to maintain a stable structure because of the de-doping phenomenon caused by light, electricity, magnetism, thermal, etc. when used as electrode materials for supercapacitors and rechargeable batteries.It is hard to balance the electrochemical performance and mechanical properties while applying PANI in electrochemical energy storage technologies including supercapacitors, rechargeable batteries and fuel cells.

As we can see, the comprehensive properties of PANI need yet to be enhanced, therefore future research should focus on development of unique nanostructures of PANI with higher surface areas and conductivities for such applications. In addition, there are yet some research gaps that should be filled: for supercapacitors and rechargeable batteries, works on designing tailor-made derivatives of PANI and functionalized PANI need to be deeply explored; as for fuel cells, reports on catalysts for oxygen evolution reaction (OER) related to PANI are lacking, it should be deeply explored.

PANI is considered as one of the useful electronic and intrinsic CPs and its applications in electrochemical energy storage and conversion field have been depicted minutely in this review. We know that PANI has many unique properties, and it is likely to be useful in other fields, so it is suggested that the application of PANI should be extended to other fields, which can greatly enlarge its range of use. For instance, instability of nano-fluids caused by the gradual sedimentation (or scale formation) of nanoparticles simultaneously with agglomerating or clustering of nanoparticles inside the base fluid [174] could be improved by PANI coating. PANI with excellent flexibility can effectively encapsulate the nanoparticles, avoiding the agglomerating or clustering of nanoparticles inside the base fluid, as a result, highly efficient heat transfer of nano-fluids would be achieved. It is suggested here PANI could be explored in the application of nano-fluids for heat transfer, that is, it deserves more research in this field, and to go a step further, it deserves more research in other novel fields. By reading this review, researchers can understand the developmental tendency of PANI in the field of electrochemical energy storage and conversion, moreover, they can be inspired to discuss and study whether PANI have potential applications in other fields.

Finally, manufacturing cost is also a crucial factor for commercialization, in summary, future advances will require continuous explorations and endeavors in designing unique nanostructures of PANI with higher surface areas and conductivities, extending application fields and developing cost-effective manufacturing technologies.

## Figures and Tables

**Figure 1 materials-13-00548-f001:**
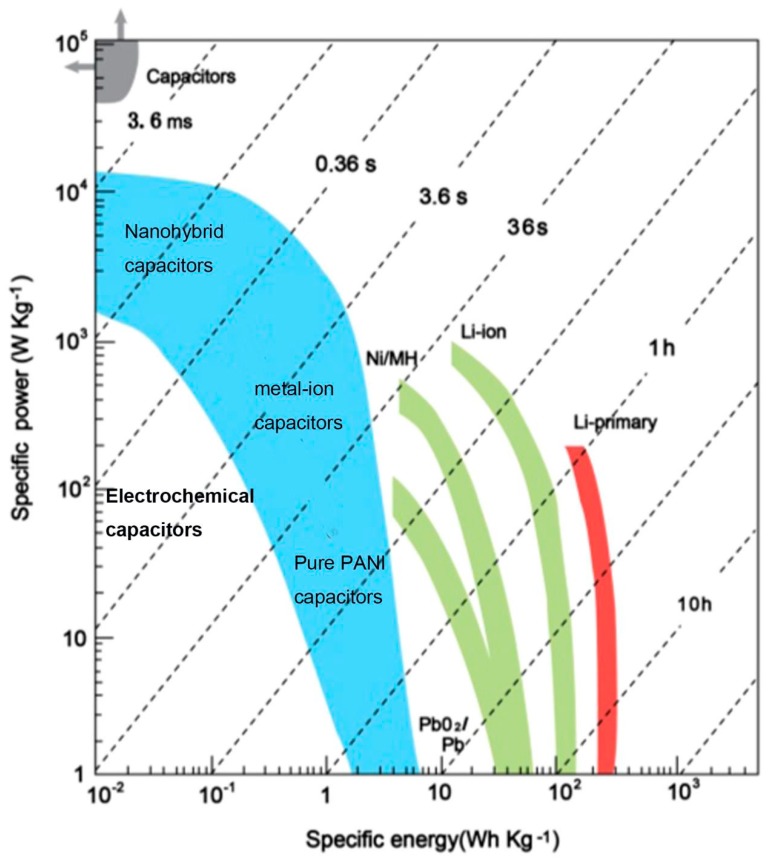
Ragone plot for various supercapacitors, batteries and fuel cells [2] (reproduced with permission from Elsevier).

**Figure 2 materials-13-00548-f002:**
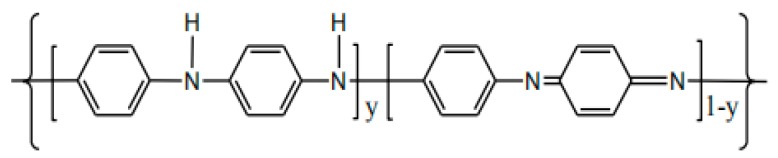
Molecular structure of polyaniline (PANI).

**Figure 3 materials-13-00548-f003:**
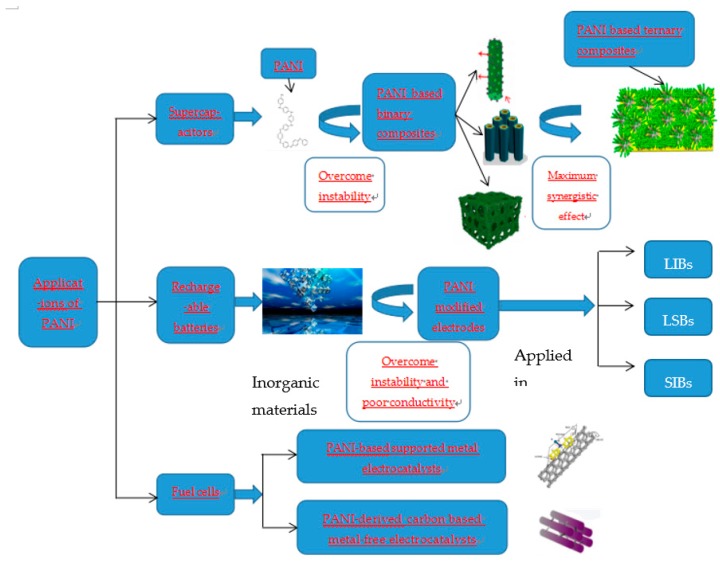
The architecture of this review.

**Figure 4 materials-13-00548-f004:**
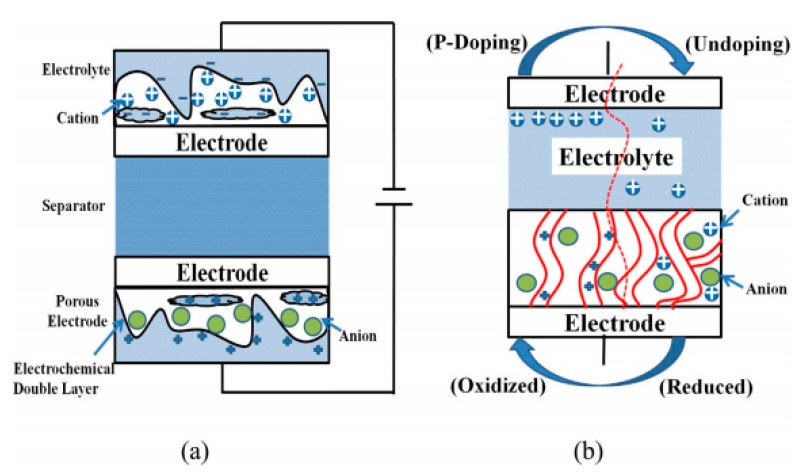
Schematics of an electrostatic double-layer capacitor (EDLC) (**a**) and pseudocapacitor; (**b**) [2] (reproduced with permission from Elsevier).

**Figure 5 materials-13-00548-f005:**
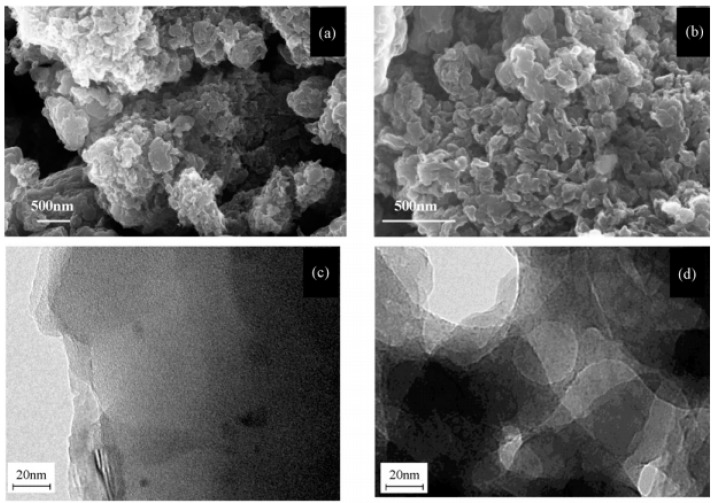
SEM images: (**a**) nonporous PANI and (**b**) porous PANI and TEM images: (**c**) nonporous PANI and (**d**) porous PANI [33] (reproduced with permission from Elsevier).

**Figure 6 materials-13-00548-f006:**
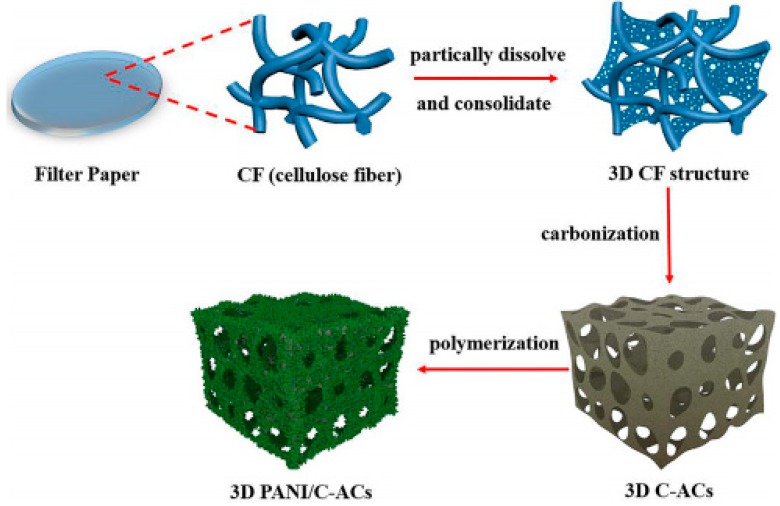
Schematic of PANI/C-ACs synthesis procedure [36] (reproduced with permission from Elsevier).

**Figure 7 materials-13-00548-f007:**
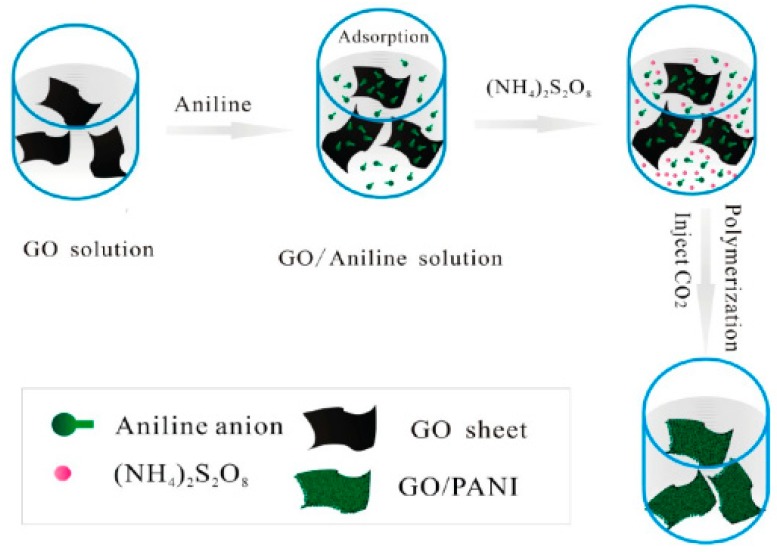
Schematic demonstration of the synthesis process of PANI/GO nanocomposite. (reprinted with permission from previous literature [50] © 2012 American Chemical Society).

**Figure 8 materials-13-00548-f008:**
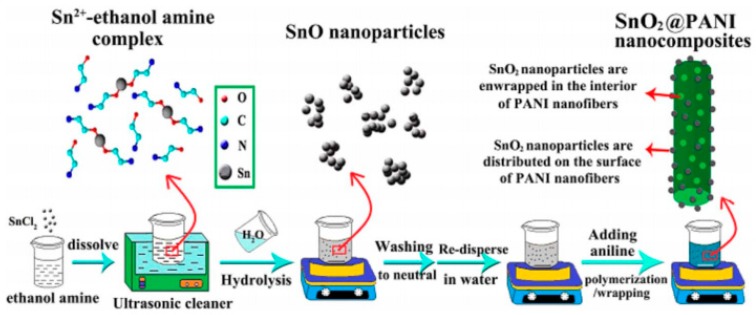
Schematic diagram of preparation of SnO and SnO_2_@PANI nanocomposite (reproduced from [67] with permission from The Royal Society of Chemistry).

**Figure 9 materials-13-00548-f009:**
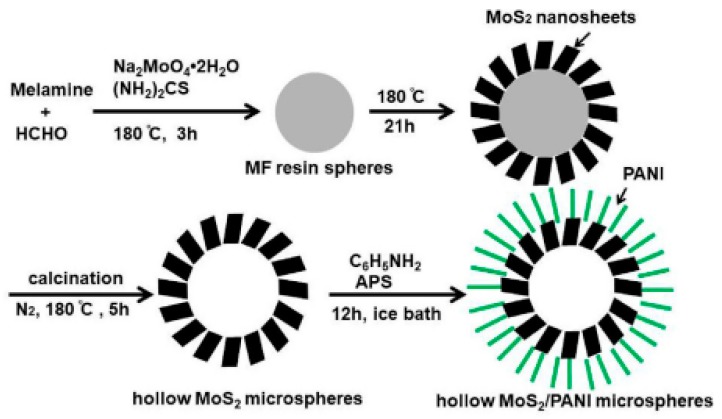
Schematic illustration of the formation of MoS_2_/PANI hollow microspheres [69] (reproduced with permission from Elsevier).

**Figure 10 materials-13-00548-f010:**
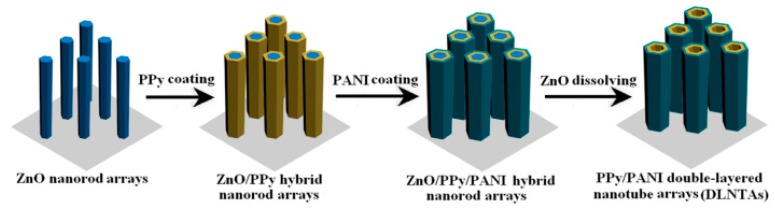
Schematic diagram of the synthesis of PANI/PPy double-walled nanotube arrays (DNTAs; reprinted with permission from previous literature [75] © 2014 American Chemical Society).

**Figure 11 materials-13-00548-f011:**
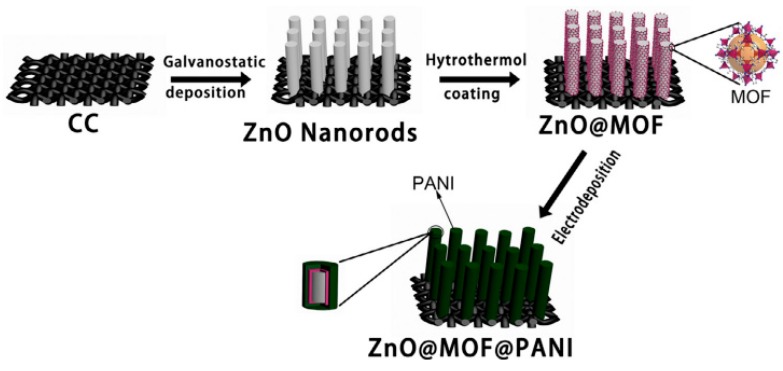
Schematic illustration of synthesis of ZnO@MOF@PANI nanoarrays on carbon cloth (CC) [84] (reproduced with permission from Elsevier).

**Figure 12 materials-13-00548-f012:**
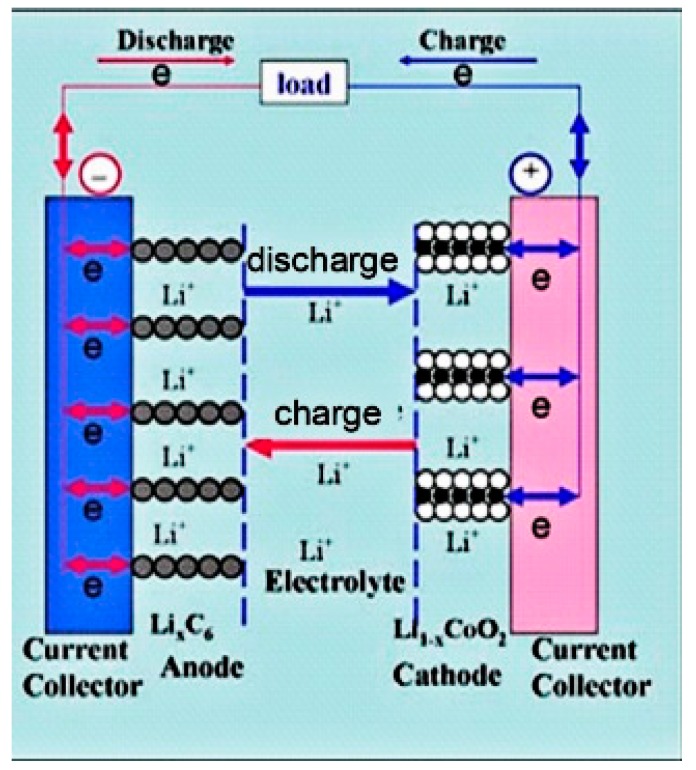
Schematic illustration of the working mechanism of LIBs.

**Figure 13 materials-13-00548-f013:**
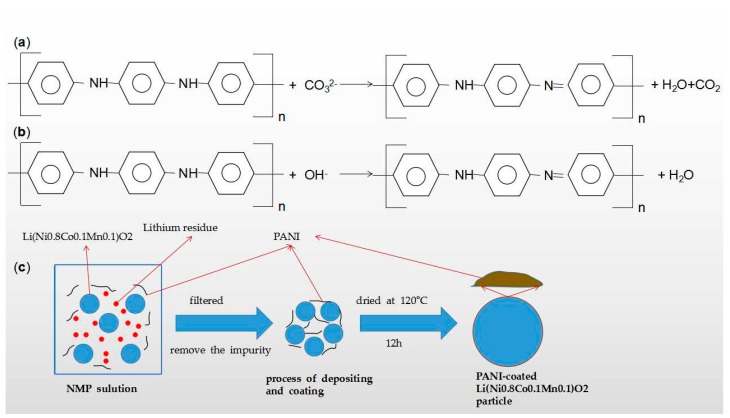
Reactions of protonated PANI with residual lithium compounds like Li_2_CO_3_ (**a**) and LiOH (**b**) and (**c**) schematic illustration of preparation of PANI-coated Li(Ni_0.8_Co_0.1_Mn_0.1_)O_2_ cathode material.

**Figure 14 materials-13-00548-f014:**
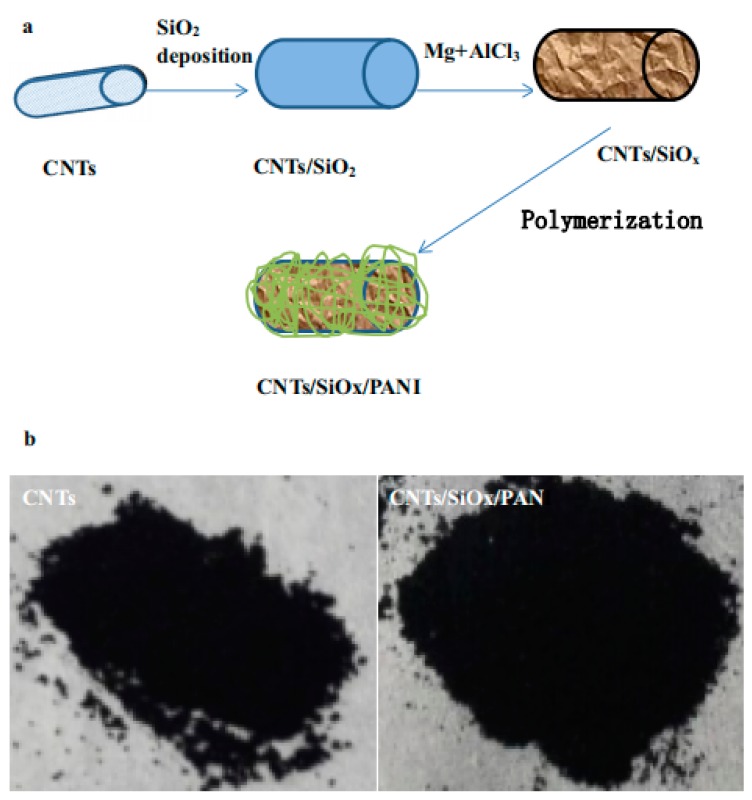
(**a**) Schematic illustration of preparation of PANI/SiO_x_/CNTs and (**b**) digital photographics of PANI/SiO_x_/CNTs [116] (reproduced with permission from Elsevier).

**Figure 15 materials-13-00548-f015:**
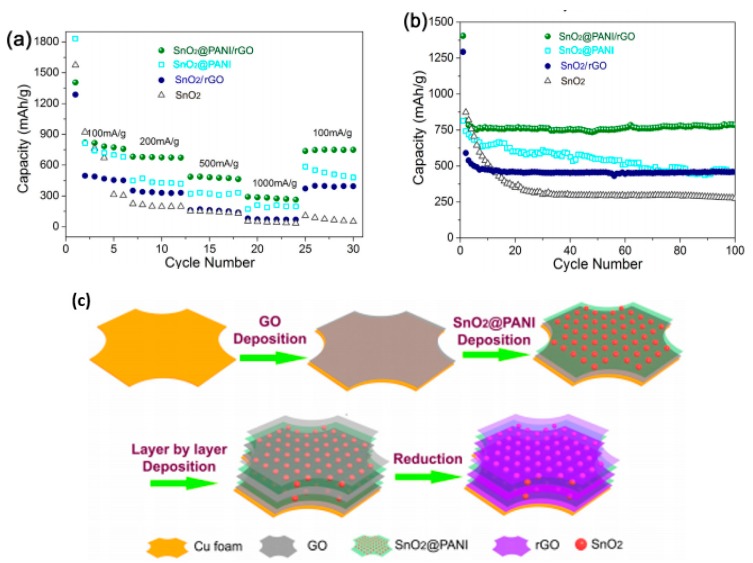
(**a**) Rate capabilities of the SnO_2_@PANI/rGO nanocomposites, SnO_2_@PANI, SnO_2_/RGO and SnO_2_ respectively; (**b**) cyclic behaviors of the SnO_2_@PANI/rGO composites, SnO_2_/rGO, SnO_2_@PANI and SnO_2_ at the current density of 100 mA/g and (**c**) a schematic illustration of the preparation of the PANI/SnO_2_/RGO nanocomposite [120] (reproduced with permission from Elsevier).

**Figure 16 materials-13-00548-f016:**
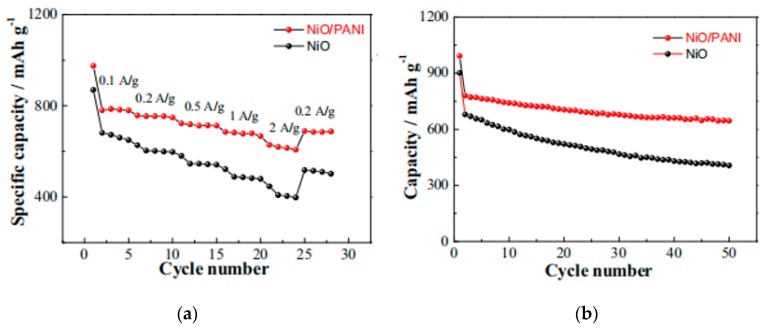
Electrochemical comparison of NiO nanoflake arrays and NiO/PANI core/shell arrays. (**a**) Rate capability and (**b**) cycling stability at 0.1 A/g [126] (reproduced with permission from Elsevier).

**Figure 17 materials-13-00548-f017:**
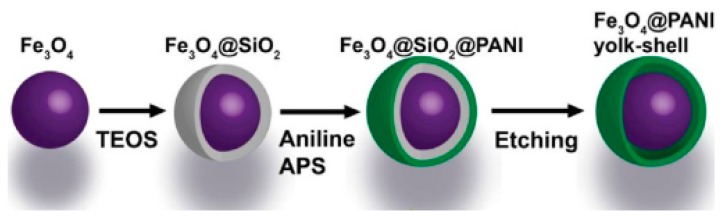
Schematic illustration of forming of Fe_3_O_4_@PANI yolk–shell micro-nanoarchitecture [128] (reproduced with permission from Elsevier).

**Figure 18 materials-13-00548-f018:**
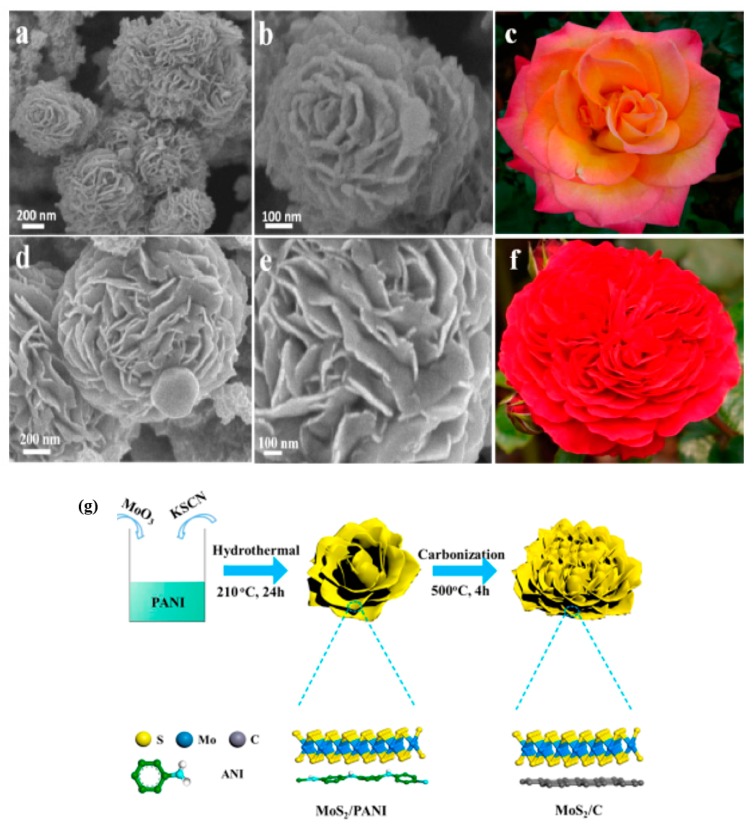
SEM images of the 3D hierarchical MoS_2_/PANI nanoflowers (**a**,**b**) and MoS_2_/C nanoflowers (**d**,**e**). Photographs of two types of Chinese roses (**c**,**f**). Schematic illustration of synthesis of MoS_2_/PANI and MoS_2_/C nanoflowers (**g**; reprinted with permission from previous literature [131] © 2014 American Chemical Society).

**Figure 19 materials-13-00548-f019:**
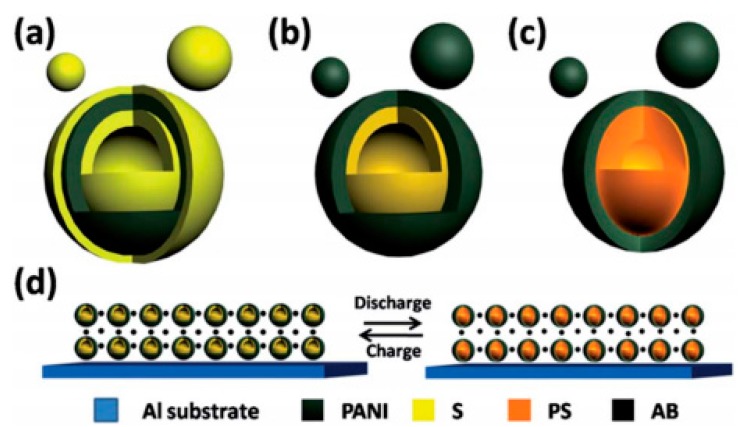
Schematic illustration of the hollow sphere PANI/S composite during the charge/discharge process. (**a**) The initial PANI-S composite, (**b**) the cycled PANI-S composite, (**c**) the lithiated PANI-S composite and (**d**) the schematic illustration of integrity of the hollow PANI-S cathode with severe volume change during charge/discharge process (reproduced from [139] with permission from The Royal Society of Chemistry).

**Figure 20 materials-13-00548-f020:**
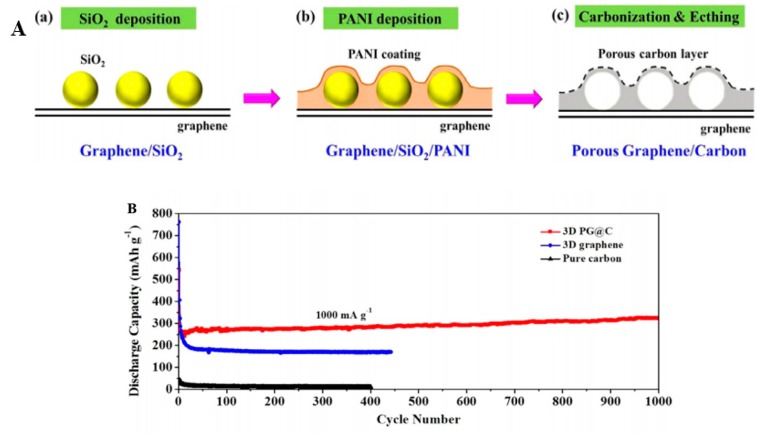
(**A**) Schematic illustration of formation of 3D porous graphene@C and (**B**) long-term cycling behavior of the 3D porous graphene@C composite at 1000 mA/g [155] (reproduced with permission from Elsevier).

**Figure 21 materials-13-00548-f021:**
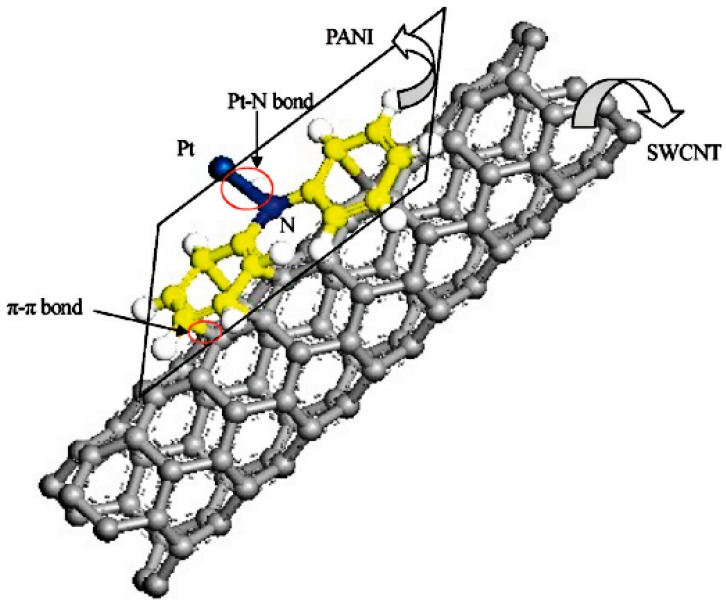
Molecular interaction in the prepared Pt-PANI/CNT catalyst (reprinted with permission from previous literature [168] © 2011 American Chemical Society).

**Figure 22 materials-13-00548-f022:**
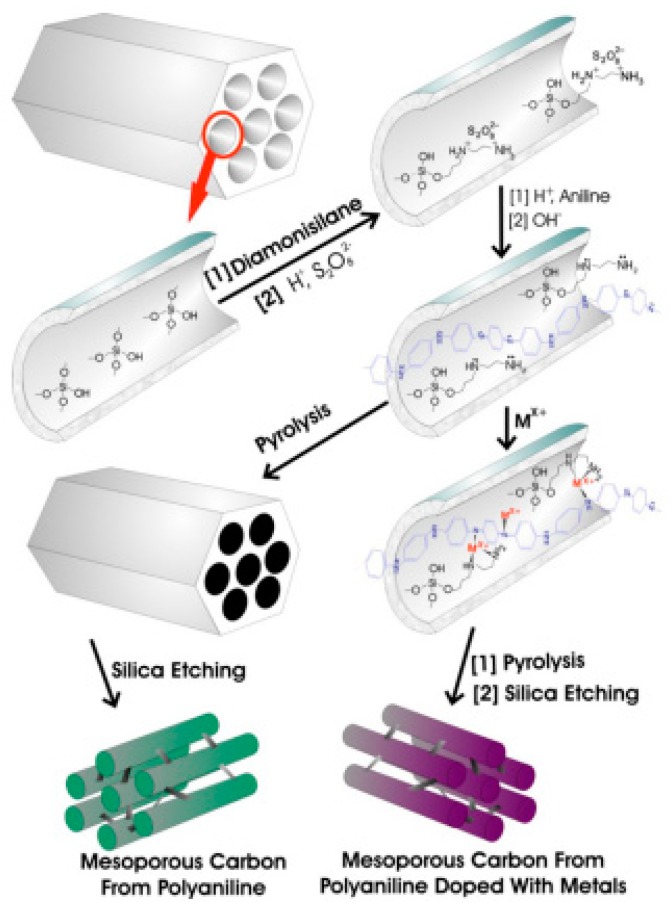
Schematic illustration of synthesis of the PDMC (reprinted with permission from previous literature [170] © 2013 American Chemical Society).

**Figure 23 materials-13-00548-f023:**
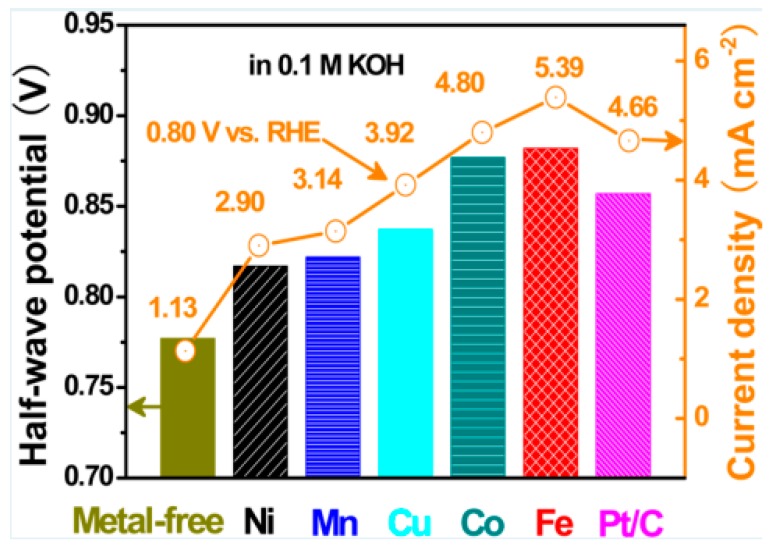
Half-wave potential vs. RHE and current density of M-PANI/C-Mela catalysts in 0.1 M KOH (reprinted with permission from previous literature [173] © 2014 American Chemical Society).

**Table 1 materials-13-00548-t001:** The preparation method and electrochemical performance of some typical PANI based supercapacitor electrode materials.

Materials	Preparation Method	Maximum Specific Capacitance	Cycle Stability
**PANI [34]**	interfacial polymerization	554 F g^−1^ at 10 mA g^−1^	57 F g^−1^ after 1000 cycles
**PANI/POROUS CARBON [35]**	electrochemical polymerization	180 F g^−1^ at 1 A g^−1^	163 F g^−1^ after 1000 cycles
**PANI/C-ACS [36]**	selective surface dissolution (SSD) method	765 F g^−1^ at 1 A/g	91% after 5000 cycles
**PANI-NWS/CMK-3 [41]**	chemical oxidative polymerization		90.4% after 1000 cycles
**NANO-HCS/PANI [43]**	in situ chemical oxidative polymerization	435 F g^−1^ at 1 A g^−1^	60% after 2000 cycles
**PANI/MWCNTS [44]**	chemical oxidative polymerization	320 F g^−1^ at 10 mA g^−1^	8% after 50 cycles
**CNTS/PANI [45]**	deposition of PANI on the surface of CNTs	183 F g^−1^ at 10 mA g^−1^	
**NITROGEN-CONTAINING CNTS/PANI [46]**	high temperature treatment	163 F g^−1^ at 700 °C and 0.1 A g^−1^	
**SWCNTS/PANI [47]**	spray-printing method	355.5 F g^−1^ at 0.1 A g^−1^	87.2% after 5000 cycles
**CCG/PANI-NFS [48]**	vacuum filtration the mixed dispersions	210 F g^−1^ at 0.3 A g^−1^	94% after 1000 cycles
**GRAPHENE/PANI NANOFIBER [49]**	in situ polymerization	480 F/g at 0.1 A/g	70% after 1000 cycles
**GO/PANI [50]**	in situ polymerization	425 F/g at 0.2 A/g	83% after 500 cycles
**GO/PANI [51]**	a soft chemical route	531 F/g at 0.2 A/g	
**RGO/PANI [52]**	in situ polymerization	1045 F/g at 0.1 A/g	97% after 1000 cycles
**PANI/CARBON NANOFIBER [53,54]**	sol–gel and electrospinning method	234 F/g at 0.1 A/g	90% after 1000 cycles
**PANI/MNO2 [60]**	pulse electrodeposition	810 F/g at 0.5 A/g	86.3% after 1000 cycles
**PANI/MNO2 NANOFIBER MICROSPHERE [61]**	interfacial chemical polymerization	765 F/g at 1 mA/cm^2^	85.1% after 400 cycles
**NANO-PANI@MNO2 [62]**	surface initiated polymerization	386 F/g with the potential window range from 0 to 0.6 V	79.5% after 800 cycles
**PANI/CUO [64]**	in situ polymerization	286.35 F/g at 20 mV/s	
**RUO2/PANI [65]**	in situ oxidative polymerization	425 F/g at 1 mA/cm^2^	
**PANI/FE3O4 [66]**	in situ polymerization	572 F/g at 0.5 A/g	82% over 5000 cycles
**SNO2/PANI [67]**	in situ oxidative polymerization	335.5 F/g at 0.1 A/g	no capacitance decay after 1000 cycles
**PANI/CUCL2 [68]**	in situ oxidative polymerization	626 F/g at 10 mV/s	
**MOS2/PANI [69]**	template-assisted technique	364 F/g at 5 mV/s	84.3% after 8000 cycles
**PANI/TIN NWAS [70]**	electrodeposition technique	1064 F/g at 1 A/g	95% after 200 cycles
**PANI/PPY DNTAS [75]**	PPy coated onto the PANI	693 F/g at 5 mV/s	92.4% over 1000 cycles
**PANI/AG [77]**	Ag nanoparticles dispersed onto the surface of PANI	553 F/g at 1 A/g	90% after 1000 cycles
**MNFE2O4/GRAPHENE/PANI [79]**	in situ chemical polymerization	241 F/g at 0.5 mA/cm^2^	100% after 5000 cycles
**3DRGN-MNO2-PANI) [80]**	electro-polymerization	1181 F/g at 1 A/g	89.1% after 1000 cycles
**PANI/RGO/CEO2 [81]**	spray drying method	684 F/g at 1 A/g	92% after 6000 cycles
**AG/MNO2/PANI [82]**	pulsed potential electro-deposition	621 F/g and 800 F/g at 1 A/g from CV and CD respectively	83% after 750 cycles
**PANI@TIO2/TI3C2TX [83]**	hydrothermal treatment with addition of in situ polymerization process	188.3 F/g at 10 mV/s	94% after 8000 cycles

**Table 2 materials-13-00548-t002:** The preparation method and electrochemical performance of some typical PANI modified rechargeable batteries electrode materials.

Materials	Preparation Method	Maximum Specific Capacity	Cycle Stability
**PANI/LICOO2 [87,88]**	Pickering emulsion route	136 mAh/g	
**PANI-CSA/C-LFP [91]**	coating C-LFP with PANI-CAS in m-cresol solution	165.3 mAh/g	
**PANI/PEG [93]**		125.3 mAh/g	95.7% after 100 cycles
**LIV3O8/PANI [94]**	oxidative polymerization		95% after 55 cycles
**PANI/LI(NI0.8CO0.1MN0.1)O2 [96]**	solution method	193.8 mAh/g	96.25% after 80 cycles
**PANI/LI(NI1/3MN1/3FE1/3)O2 [97]**			86% after 40 cycles
**PANI [101]**	chemical oxidative polymerization	159.83 mAh/g	119.79 mAh/g retained after 100 cycles
**PANI/SI/GRAPHITE [108]**		1392 mAh/g	62.2% after 95 cycles
**CNTS/PANI/SI [113]**		1954 mAh/g	727 mAh/g retained after 100 cycles
**PANI/SIOX/CNTS [116]**		1156 mAh/g	728 mAh/g retained after 60 cycles
**SIOX/PANI/CU2O [117,118]**		1178 mAh/g	725 mAh/g retained after 60 cycles
**PANI/SNO2/RGO [120]**	dip-coating of PANI@SnO_2_ and graphene dispersion on Cu foam	772 mAh/g	749 mAh/g retained after 100 cycles
**EG/PANI /SNO2 [121]**	solvothermal method followed by in-situ oxidative polymerization	1021 mAh/g	408 mAh/g retained after 100 cycles
**PANI@SNO2@MWCNT [122]**		888 mAh/g	527 mAh/g retained after 150 cycles
**TIO2/PANI/GO [124]**		1335 mAh/g	435 mAh/g after 250 cycles
PANI/FE2O3 [127]	solvothermal technique followed by a post-coating process	366 mAh/g at 2.0 C	412.1 mAh/g retained after 150 cycles at 0.2 C
FE3O4@PANI [128]		997 mAh/g	982 mAh/g retained after 50 cycles
**GRAPHENE/FE3O4/PANI [129]**		1214 mAh/g	86% after 50 cycles
SNS2@PANI [132]		968.7 mAh/g	75.4% after 80 cycles
**NANO-POROUS SULFUR/PANI [137]**	in situ chlorinated substitution and vulcanization reactions	750 mAh/g	89.7% after 200 cycles
**CPNRS/FEPO4 [147]**	microemulsion method	140.2 mAh/g	134.4 mAh/g retained after 120 cycles
**NVP@C@HC [148]**			83.3% after 3000 cycles
**S/N-CT [156]**		340 mAh/g	94% after 3000 cycles

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
