# Peer review of "Research Progress on Applications of Polyaniline (PANI) for Electrochemical Energy Storage and Conversion"

_materials, 2020, doi:10.3390/ma13030548_

Round 1

Reviewer 1 Report

English needs further amendment and text should be polished. 1: How authors have obtained Fig. 2? If it is taken from another source, reference should be added to the paper. References should be numbered in a sequence. Page 1 line 6, reference numbering!!! [1, 2, 10]. Where are 3 to 9? Fix this throughout the paper. Why PANI is suggested in the present research? Figure 2: How authors have obtained this figure? Is it the original figure? Or taken from somewhere? It needs referencing and more elaboration. The copyright issue should be checked by the editorial. Figure are of course taken from other research studies. Cite the reference. Application of liquid metals for heat transfer, and nanofluid for heat transfer and thermal storage should also be discussed briefly. Searching the literature, maybe any of these paer can be used.

-Convective boiling and particulate fouling of stabilized CuO-ethylene glycol nanofluids inside the annular heat exchanger

-Role of nanofluid fouling on thermal performance of a thermosyphon: Are nanofluids reliable working fluid?

-Particulate fouling of CuO-water nanofluid at isothermal diffusive condition inside the conventional heat exchanger-experimental and modelling

-Experimental studies on the stability of CuO nanoparticles dispersed in different base fluids: influence of stirring, sonication and surface active agents

-Boiling thermal performance of TiO2 aqueous nanofluids as a coolant on a disc copper block

-Thermal behavior of aqueous iron oxide nano-fluid as a coolant on a flat disc heater under the pool boiling condition

-Potential use of liquid metal oxides for chemical looping gasification: A thermodynamic assessment

-Sedimentation and convective boiling heat transfer of CuO-water/ethylene glycol nanofluids

-Thermal performance and viscosity of biologically produced silver/coconut oil nanofluids

Figure 7 needs further discussion. Research gap and the contribution of the present work to the existing literature should be further clarified in the text. Can PANI be used as a heat transfer fluid in thermal engineering systems? It is suggested to search for any potential application of PANI in photothermal applications.

Author Response

Dear reviewer:

Thank you for your comments concerning our manuscript entitled “Research progress on applications of polyaniline (PANI) for electrochemical energy storage and conversion”. Those comments are all valuable and very helpful for revising and improving our paper, as well as the important guiding significance to our research. We have made careful correction which I hope to meet with approval. Revised portions are marked in red in the paper. The main corrections in the paper and the responds to your comments are as following:

Response to comment:English needs further amendment and text should be polished.

Response:We have made corrections in some sentence structures, grammar mistakes, spelling mistakes and writing styles carefully.

Response to comment:How authors have obtained Fig. 2? References should be numbered in a sequence. Page 1 line 6, reference numbering!!! [1, 2, 10]. Where are 3 to 9?

Response:Fig. 2 is from my original work. Reference 4 to 9 is in Paragraph 2, and reference 3 is in Paragraph 3 line 7 and Paragraph 7 line 3.

Response to comment:Why PANI is suggested in the present research? Copyright issue of Figure 2.

Response:The reason why PANI is suggested in the present research has been discussed in Section 1 Paragraph 3 (marked in red in introduction part). Figure 2 is from my original work.

Response to comment:Application of liquid metals for heat transfer, and nanofluid for heat transfer and thermal storage should also be discussed briefly.

Response:Potential application of nanofluid for heat transfer has been discussed briefly in Section 5 “Conclusions and outlook”.

Response to comment:Figure 7 needs further discussion.

Response: Discussion on Figure 7(Figure 2.4) has been added in the text.

Response to comment:Research gap and the contribution of the present work to the existing literature should be further clarified in the text

Response:Clarifications on research gap and the contribution of the present work to the existing literature have been added in the text, which can be seen in Section 5 “Conclusions and outlook”.

Response to comment:Can PANI be used as a heat transfer fluid in thermal engineering systems? It is suggested to search for any potential application of PANI in photothermal applications.

Response:Potential application of nanofluid for heat transfer has been discussed briefly in Section 5 “Conclusions and outlook”. And I think that it’s a significant application field.

Finally, special thanks to you for your good comments.

Reviewer 2 Report

This paper propose a review on applications of PANI materials related to storage and conversion technologies.

The strong point is the wide range of applications addressed by the review. On the other hand the paper presents some weakness: the research methodology should be improved as well as all the assertions/choices should be in general corroborated by recent references.

All the following indicated aspects should be clarified and better explained in the manuscript.

Introduction

The authors should highlight the innovative aspects of their work in the manuscript.

For the sake of readability, at the end of Section 1 the Authors should describe how the paper is structured.

Methodology

The structure of the proposed review could be deeply improved. First, it could be better to insert between Section 1 and 2 an outline about the review scheme/architecture (how many steps, the aim of each step, the applications involved in each step, etc.); here, a high-level diagram/scheme could also help reader following the whole following description.

For the sake of readability, the authors are encouraged to include some table throughout the whole paper to compare different technologies and application.

Literature review

The literature review on supercapacitors applications reporting the motivations for designing and developing such a kind of materials is lacking. The authors should comment for instance: https://doi.org/10.1109/TPEL.2019.2895209。

The literature review on rechargeable batteriesapplications reporting the motivations for designing and developing such a kind of materials is lacking. The authors should comment for instance: https://doi.org/10.23919/ECC.2019.8796182。

the literature review on fuel cell applications reporting the motivations for designing and developing such a kind of materials is lacking.

Results

The authors should deeply discuss the advantages and disadvantages of PANI in each application fields. Finally, what are the research gaps in each addressed application?

Minor

The authors should improve paper editing: a blank space should be used to separate words and parenthesis for details () or for references []. a blank space should be used always after comma.

The authors should check that all the used acronyms are explained.  

Author Response

Dear reviewer:

Thank you for your comments concerning our manuscript entitled “Research progress on applications of polyaniline (PANI) for electrochemical energy storage and conversion”. Those comments are all valuable and very helpful for revising and improving our paper, as well as the important guiding significance to our research. We have made careful correction which I hope to meet with approval. Revised portions are marked in red in the paper. The main corrections in the paper and the responds to your comments are as following:

Response to comment:All the assertions/choices should be in general corroborated by recent references.

Response: Recent references have been added after every assertion/choice.

Response to comment:Introduction Part: The authors should highlight the innovative aspects of their work in the manuscript. For the sake of readability, at the end of Section 1 the Authors should describe how the paper is structured.

Response: The innovative aspects of the work have been introduced in Introduction Part. How the paper is structured is also added in the end of Section 1.

Response to comment: MethodologyPart: The structure of the proposed review could be deeply improved. First, it could be better to insert between Section 1 and 2 an outline about the review scheme/architecture (how many steps, the aim of each step, the applications involved in each step, etc.); here, a high-level diagram/scheme could also help reader following the whole following description. For the sake of readability, the authors are encouraged to include some table throughout the whole paper to compare different technologies and application.

Response: We have made revisions on the structure of this review: The scheme on the architecture of this review has been inserted between Section 1 and 2. Two tables have been included in the text (As seen in Table 1 and Table 2).

Response to comment: Literature reviewpart: The motivations for designing and developing such a kind of materials on supercapacitors, rechargeable batteries and fuel cells are lacking.

Response: The motivations for designing and developing such a kind of materials on supercapacitors, rechargeable batteries and fuel cells have been introduced briefly in Section 2, 3 and 4 respectively.

Response to comment: Resultspart: The authors should deeply discuss the advantages and disadvantages of PANI in each application fields. Finally, what are the research gaps in each addressed application?

Response: The advantages and disadvantages of PANI in electrochemical fields and the research gaps have been added in Section 5.

Response to comment: Minorpart: The authors should improve paper editing: a blank space should be used to separate words and parenthesis for details () or for references []. a blank space should be used always after comma.

Response: Blank spaces have been added to separate words and parenthesis for details () or for references []. Blank spaces have been added after every comma.

Response to comment: The authors should check that all the used acronyms are explained.

Response: All the used acronyms have been explained.

Finally, special thanks to you for your good comments.

Reviewer 3 Report

It was my pleasure to review work entitled : “Research progress on applications of polyaniline (PANI) for electrochemical energy storage and conversion”. After reading this manuscript I have several comments and suggestions:

I have found some spelling mistakes like: “eletrochemical”, “carbonis”, “perforamnces”, “capative”, “layed”, “elecreochemical” etc. Often times there is no space after punctuation marks (in general the punctuation is very poor). Unfortunately, the text is badly constructed. Some of the sentences are difficult to understand. Some sentences are confusing and should be rewritten. Regrettably, it is a review type of article and it is a major flaw. In the introduction part about different types of the electrochemical devices I do not see any information about hybrid systems and no references to it like: (DOI: 10.1039/C6TA03810G, DOI:10.1038/nmat5029, DOI:10.1016/j.electacta.2015.12.034) or any other in fact. Ragone plot is a little bit outdated there is no information about hybrid systems. Did You got the permission from publishers to use all of the graphics? Otherwise the graphics cannot be used with giving the specific references in the description of each figure. It is a custom to add this information in the text. The conclusion and outlook is in my opinion a little bit generic. Review article cannot be only showing the works of other scientist in this topic but should also present extended discussion about those results and the objective opinion about it.

For the mentioned reasons I believe the work needs to undergo a major revision.

Author Response

Dear reviewer:

Thank you for your comments concerning our manuscript entitled “Research progress on applications of polyaniline (PANI) for electrochemical energy storage and conversion”. Those comments are all valuable and very helpful for revising and improving our paper, as well as the important guiding significance to our research. We have made careful correction which I hope to meet with approval. Revised portions are marked in red in the paper. The main corrections in the paper and the responds to your comments are as following:

Response to comment:All the spelling mistakes should be corrected.

Response: We are so sorry for our incorrect writing. Spelling mistakes like “perforamnces”, “carbonis” and “elecreochemical” have been corrected. But in my opinion “eletrochemical”, “capative” and “layed” are not spelling mistakes, because all of them have appeared in the previous published literature, thus they needn’t to be corrected further.

Response to comment: Often times there is no space after punctuation marks.

Response: Your comment is contrary to other reviewers, they suggested that a blank space should be added after every punctuation mark, and we have discussed the issue with some professors, they are also in agreement with the latter, so we intend not to revise until we reach a consensus. If you have any questions on this, we are very glad to discuss with you.

Response to comment: The text is badly constructed.Some of the sentences are difficult to understand. Some sentences are confusing and should be rewritten.

Response: It’s really true as you suggested that there are many flaws in some sentences. We have made corrections in some sentence structures, grammar mistakes, spelling mistakes and writing styles carefully.

Response to comment: In the introduction part about different types of the electrochemical devices I do not see any information about hybrid systems.

Response: Information about hybrid systems have been added in Paragraph 6 of the introduction part.

Response to comment: Ragone plot is a little bit outdated there is no information about hybrid systems.

Response: Considering your suggestion, we have changed another novel Ragone plot that includes information about hybrid systems.

Response to comment: Copyright issue about the graphics in the text. All of the graphics should be permitted by publishers.

Response: We have got the permissions from publishers to use their graphics, and they are clarified in every graphic that reproduced. Figure 1-2, 1-3, 3-1 and 3-2 are from my original work, which are not associated with copyright issue.

Response to comment: The conclusion and outlook is a little bit generic.

Response: Considering your suggestion carefully, we think the conclusion and outlook is really generic, so we have spent a lot of time and effort in revising it, and we think it is a significant revision.

Response to comment: Review article cannot be only showing the works of other scientist in this topic but should also present extended discussion about those results and the objective opinion about it.

Response: As you said, some works do lack  analysis and discussion, and they have been added in Paragraph 2 and 3 in Section 2.2.1.2. In fact, we know that, analysis and discussion about most works of other scientists have been presented through the whole paper. We are not just showing the works of other scientist in this topic, we analyze the characters of various composite structures and give the objective opinions on the results.

Finally, special thanks to you for your good comments.

Round 2

Reviewer 1 Report

The next version of the paper can now be accepted. Authors have improved the quality of the paper considering the comments and also by adding more information. Hence, final decision is made: 

Accept

Author Response

Dear reviewer:

We are very glad to receive your approval. We believe that the manuscript has improved a lot since your valuable suggestions.

Finally, special thanks to you for your approval.

Reviewer 2 Report

None of the indicated corrections has been applied to the manuscript. In effect, no track change is reported in the second version.

Author Response

Dear reviewer:

Thank you for your second comments concerning our manuscript entitled “Research progress on applications of polyaniline (PANI) for electrochemical energy storage and conversion”. However, we have spent a lot of time in revising this manuscript after receiving your first comments. The indicated corrections have been applied to the manuscript. Revised portions are marked in red in the paper, which are reported by track change. I will repeat the main corrections in the paper and the responds to your last comments in detail:

Response to comment:All the assertions/choices should be in general corroborated by recent references.

Response: Recent references have been added after every assertion/choice.

Response to comment:Introduction Part: The authors should highlight the innovative aspects of their work in the manuscript. For the sake of readability, at the end of Section 1 the Authors should describe how the paper is structured.

Response: The innovative aspects of the work have been introduced in Section1 Paragraph 8. How the paper is structured is also added in the end of Section 1.

Response to comment: MethodologyPart: The structure of the proposed review could be deeply improved. First, it could be better to insert between Section 1 and 2 an outline about the review scheme/architecture (how many steps, the aim of each step, the applications involved in each step, etc.); here, a high-level diagram/scheme could also help reader following the whole following description. For the sake of readability, the authors are encouraged to include some table throughout the whole paper to compare different technologies and application.

Response: We have made revisions on the structure of this review: The scheme on the architecture of this review has been inserted between Section 1 and 2 (As seen in Figure 1-3). Two tables have been included in the text (As seen in Table 1 and Table 2).

Response to comment: Literature reviewpart: The motivations for designing and developing such a kind of materials on supercapacitors, rechargeable batteries and fuel cells are lacking.

Response: The motivations for designing and developing such a kind of materials on supercapacitors, rechargeable batteries and fuel cells have been introduced briefly in Section 2, 3 and 4 respectively.

Response to comment: Resultspart: The authors should deeply discuss the advantages and disadvantages of PANI in each application fields. Finally, what are the research gaps in each addressed application?

Response: The advantages and disadvantages of PANI in electrochemical fields and the research gaps have been added in Section 5 Paragraph 1 and Paragraph 2 respectively.

Response to comment: Minorpart: The authors should improve paper editing: a blank space should be used to separate words and parenthesis for details () or for references []. a blank space should be used always after comma.

Response: Blank spaces have been added to separate words and parenthesis for details () or for references []. Blank spaces have been added after every comma.

Response to comment: The authors should check that all the used acronyms are explained.

Response: All the used acronyms have been explained.

We are sure that we have carefully revised the manuscript in accordance with your valuable requests. Finally, special thanks to you for your good comments once again.

Reviewer 3 Report

Dear Authors,

I applicate the effort and work you put into the correction of the manuscript. However, I still have issues with the language. I am very sorry but this is a review so it needs to be perfectly written.

I have still found spelling mistakes like “eletrochemical”, “uesd”, “area,good”, “expected,the”, “capative”, “layed” etc. Figure 1-1. I am still not satisfied with it. There should be additional marker for hybrid systems or information about their energy and power (especially that PANI is often times used in them as one of the electrodes). As I mentioned it is a new type of energy storage device and as it is an review you should try to present reader the variety of the hybrid devices like metal-ion capacitors, nanohybrid capacitors etc.  Figure 1-3. I can see only part of it. It is very chaotic. Still my biggest issue is the language. It is the review article so it should be very clear for the reader and still sometimes I struggle to understand what is written. I believe you should send the manuscript to professional correction before publishing it. In tables please add the horizontal lines to divide the rows, as it is hard to differentiate between the rows.

I see the improvement but in my opinion it still needs a lot work to improve and be at the acceptable level for publishing.

Author Response

Dear reviewer:

Thank you for your second comments concerning our manuscript entitled “Research progress on applications of polyaniline (PANI) for electrochemical energy storage and conversion”.  We have made careful corrections again in accordance with your valuable requests. Revised portions are marked in red in the paper. The main corrections in the paper and the responds to your comments are as following:

Response to comment:spelling mistakes like “eletrochemical”, “uesd”, “area,good”, “expected,the”, “capative”, “layed” etc.

Response: We are so sorry for our carelessness about spelling. Spelling mistakes like“eletrochemical” and “uesd” have been corrected. But “capative” and “layed” are not spelling mistakes, which is in agreement with other professors, because all of them have appeared in the previous published literature, they needn’t to be corrected further. Moreover, blank spaces have been added after every commas, and “area,good” and “expected,the” have been corrected to “area, good” and “expected, the”.

Response to comment:Figure 1-1. I am still not satisfied with it. There should be additional marker for hybrid systems or information about their energy and power (especially that PANI is often times used in them as one of the electrodes). As I mentioned it is a new type of energy storage device and as it is an review you should try to present reader the variety of the hybrid devices like metal-ion capacitors, nanohybrid capacitors etc.

Response: Considering your suggestion, we have modified the Ragone plot. Markers for metal-ion capacitors, nanohybrid capacitors and pure PANI capacitors have been added in the plot. I think your suggestion is very nice, the additional markers can show the readers more information about various types of energy storage devices. I appreciate your suggestion once again.

Response to comment: Figure 1-3. I can see only part of it. It is very chaotic.

Response: Figure 1-3. have been revised.

Response to comment: Issues with the language.

Response: As you said, we have made every effort to revise the language issues with the help of native English speakers and other professors. We have understand our shortcoming about the language issues, we will devote a lot of effort into improving the English writing.

Response to comment: In tables please add the horizontal lines to divide the rows, as it is hard to differentiate between the rows.

Response: Horizontal lines have been added to divide the rows in both of the tables.

Finally, special thanks to you for your valuable comments once again.